# A potent small-molecule inhibitor of the DCN1-UBC12 interaction that selectively blocks cullin 3 neddylation

Haibin Zhou[1], Jianfeng Lu[1], Liu Liu[1], Denzil Bernard[1], Chao-Yie Yang[1], Ester Fernandez-Salas[2], Krishnapriya Chinnaswamy[3], Stephanie Layton[3], Jeanne Stuckey[3], Qing Yu[6], Weihua Zhou[4], Zhenqiang Pan[5], Yi Sun[4,6] & Shaomeng Wang[1,7,8]

The Cullin-RING E3 ubiquitin ligases (CRLs) regulate homeostasis of ~20% of cellular proteins and their activation require neddylation of their cullin subunit. Cullin neddylation is modulated by a scaffolding DCN protein through interactions with both the cullin protein and an E2 enzyme such as UBC12. Here we report the development of DI-591 as a high-affinity, cell-permeable small-molecule inhibitor of the DCN1–UBC12 interaction. DI-591 binds to purified recombinant human DCN1 and DCN2 proteins with $K_i$ values of 10–12 nM, and disrupts the DCN1–UBC12 interaction in cells. Treatment with DI-591 selectively converts cellular cullin 3 into an un-neddylated inactive form with no or minimum effect on other cullin members. Our data firmly establish a previously unrecognized specific role of the DCN1–UBC12 interaction for cellular neddylation of cullin 3. DI-591 is an excellent probe compound to investigate the role of the cullin 3 CRL ligase in biological processes and human diseases.

[1] Department of Internal Medicine, University of Michigan, Ann Arbor, Michigan 48109, USA. [2] Department of Pathology, University of Michigan, Ann Arbor, Michigan 48109, USA. [3] Life Sciences Institute, University of Michigan, Ann Arbor, Michigan 48109, USA. [4] Division of Radiation and Cancer Biology, Department of Radiation Oncology, University of Michigan, Ann Arbor, Michigan 48109, USA. [5] Department of Oncological Sciences, Mount Sinai School of Medicine, 1425 Madison Avenue, New York, New York 10029, USA. [6] Institute of Translational Medicine, Zhejiang University School of Medicine, Hangzhou, 310029, China. [7] Department of Pharmacology, University of Michigan, Ann Arbor, Michigan 48109, USA. [8] Department of Medicinal Chemistry, University of Michigan, Ann Arbor, Michigan 48109, USA. Haibin Zhou and Jianfeng Lu contributed equally to this work. Correspondence and requests for materials should be addressed to S.W. (email: shaomeng@umich.edu)

The regulated destruction of intracellular proteins is controlled via the ubiquitin proteasome system (UPS) by covalent modification of target proteins with ubiquitin, and is essential for cellular protein homeostasis[1,2]. The cullin RING ligases (CRLs), a central component of the UPS, regulate the turnover of ~20% of cellular proteins and play a critical role in normal cellular physiology and various human diseases[3–6]. The classification of CRLs is based on the key individual cullin protein in the CRL system. Mammalian cells have eight cullins (cullin 1, 2, 3, 4A, 4B, 5, 7 and 9)[4] with cullin 1 being the most studied[7,8].

The activities of CRLs are controlled by the neural precursor cell expressed developmentally downregulated-8 (NEDD8), an ubiquitin-like protein[9–12]. Neddylation is analogous to the process of ubiquitination, wherein the ubiquitin-like protein NEDD8 is conjugated to its target proteins. The neddylation cascade begins with the activation of NEDD8 by the NEDD8-activating enzyme (NAE), an E1 enzyme, followed by the transfer of the activated NEDD8 to one of two NEDD8-specific E2 enzymes, UBC12 and UBE2F. In the final step of this cascade, E3 enzymes catalyze the transfer of NEDD8 from E2 to target substrates, with cullins being the best-characterized substrates. Neddylation of cullins results in the activation of CRLs. The structural and biochemical mechanisms underlying the E1–E2–E3 cascade reaction in the NEDD8 pathway have previously been elegantly elucidated[13–18].

The NEDD8 pathway has been pursued for the development of novel therapeutics[7,8,19,20]. MLN4924, an inhibitor of the E1 enzyme NAE, was shown to suppress tumor cell growth both in vitro and in vivo[21]. Mechanistically, MLN4924 inhibits NAE enzymatic activity through formation of a covalent NEDD8–MLN4924 adduct, blocking activation of all CRLs, and causing accumulation of CRL substrates[21,22]. MLN4924 is currently being tested in clinical trials for the treatment of human cancers[23].

Since MLN4924 inhibits the NAE E1 activity, it is very effective in achieving broad inhibition of all CRLs. Consequently, it will be invaluable to develop small molecules that can selectively modulate neddylation of individual CRL members. To this end, we have investigated a general strategy of disrupting the protein–protein interaction network within the multi-protein cullin complexes as a means to selectively modulate the activity of specific CRLs.

In the best studied cullin system, cullin 1 forms a complex with UBC12, defective in cullin neddylation 1 (DCN1), RING-box protein 1 (RBX1) and NEDD8 proteins for efficient transfer of NEDD8 from UBC12 to cullin 1[18]. DCN1 was shown to interact through two separate domains with both UBC12 and cullin 1, enhancing cullin neddylation[14–18]. Our analysis of the crystal structure of DCN1 complexed with UBC12 showed that the DCN1–UBC12 interaction appears to be amenable for the design of small-molecule inhibitors[17,18].

In the present study, we report the discovery through structure-based design and extensive medicinal chemistry optimization, of DI-591 as a high-affinity, cell-permeable, drug-like small-molecule inhibitor of the DCN1–UBC12 interaction. Employing DI-591 as a chemical probe, we demonstrate that in cells, DCN1 is surprisingly much more critical in neddylation of cullin 3 than in neddylation of cullin 1 and other cullin family members. Treatment of cells with DI-591 selectively converts cellular cullin 3 into an un-neddylated inactive form with no or minimum effect on other cullins. Inhibition of cullin 3 neddylation leads to upregulation of NRF2, a known cullin 3 CRL substrate[24–26]. DI-591 is an excellent probe compound to investigate the role of cullin 3 in biological processes and human diseases.

## Results

### Determination of a minimum UBC12 binding motif to DCN1.
The co-crystal structure of UBC12 complexed with DCN1 shows that their interaction is mediated primarily by a well-defined binding groove in DCN1 and an N-terminal 12-residue peptide of the UBC12 protein (Fig. 1a)[17,18].

To assess the feasibility of designing potent small-molecule inhibitors of the DCN1–UBC12 interaction, we performed a computational analysis of the DCN1-binding groove using a co-solvent mapping method[27–29]. This analysis identified a number of hydrophobic hotspots covering the regions occupied by residues Met1, Ile2, and Leu4 of UBC12 (Fig. 1b) and also suggested the feasibility of designing potent small-molecule inhibitors capable of blocking the DCN1–UBC12 interaction.

We attempted to identify the shortest UBC12 peptide with significant affinity to DCN1. In our optimized fluorescence polarization (FP)-based, competition-binding assay, the 12-residue UBC12 peptide (1) binds to DCN1 with a $K_i$ value of 2.6 μM, similar to the $K_d$ value reported previously for this peptide[22]. We synthesized a series of shorter peptides based upon the UBC12 peptide (1) by sequential truncation of C-terminal residues to assess the contribution of each residue to DCN1 binding (Fig. 1c). Truncation of Lys12 generated 2 that has a binding affinity to DCN1 similar to that of 1. Sequential removal of the next three residues resulted in 3, 4, and 5, which bind to DCN1 with $K_i$ values of 6.0–9.2 μM and are 2–3 times less potent than 1. Removal of Lys8 and Leu7 from 5 yielded 6 and 7, respectively, which show a 3-fold progressive loss in the binding affinity to DCN1. Deletion of Ser6 from 7 led to 8, which is equipotent with 7. Removal of Phe5 from 8 yielded a four-residue peptide 9, which has a $K_i$ value of 50 μM and is as potent as the seven-residue peptide (6). Deletion of Leu4 from 9 resulted in a tripeptide (10), showing no appreciable binding to DCN1 at concentrations up to 1,000 μM. We therefore used the tetrapeptide (9) as a starting point for subsequent structure-based optimization.

### Discovery of DI-591 through structure-based optimization.
The co-crystal structure of UBC12 complexed with DCN1 shows that the side chain of Met1 of UBC12 is inserted deeply into a hydrophobic pocket in DCN1 and our computational co-solvent mapping analysis suggested that the interactions at this site can be further optimized. We, therefore, performed extensive modifications of the Met1 side chain of peptide 9 (Fig. 1d). First, we synthesized a series of peptides by replacing the Met1 residue with commercially available amino acids. Replacement of Met1 with phenylalanine for example, yielded 11 whose binding affinity is similar to that of 9. However, replacement of Met1 with a homophenylalanine, led to 12, whose binding affinity was decreased by a factor of 6. We next explored the effect of different substituents on the phenyl group in 11. Substitution of this phenyl ring with a Cl atom at the ortho, meta, or para positions yielded 13, 14, and 15, respectively. None of these Cl substitutions improved the binding affinity over that of 11 notwithstanding the hydrophobic nature of the binding pocket. In fact, the o-Cl substitution reduces the binding affinity by a factor of 3 but Cl at the meta position does not decrease the binding affinity, and therefore we tested compounds containing m-CN or m-CF₃ substituents phenyl and found that both substitutions result in compounds with no appreciable binding at 1,000 μM. Other substitutions such as p-OH, o-F, and 3,5-di-F also failed to improve the binding affinity. Our modeling suggested that the binding pocket occupied by the Met1 side chain is large enough to accommodate a bicyclic aromatic ring and thus four compounds containing different bicyclic aromatic ring systems were

synthesized and evaluated. Replacement of the phenyl in **11** with indol-3-yl resulted in **21** that, with a $K_i$ value of 82 μM, is slightly less potent than **11**. Replacement of the phenyl with naphthalen-1-yl resulted in **22** which, with $K_i = 354$ μM, is nine times less potent than **11**. However, replacement of the phenyl with naphthalen-2-yl group yielded **23**, which is 11 times more potent than **11**. Replacing the phenyl group with a benzothiazol-2-yl group produced **24** that has a $K_i$ value of 150 nM and is >200 times more potent than **11**. Our modeled structure of **24** in a complex with DCN1 suggested that there is additional space around its benzothiazol-2-yl group. Accordingly, we synthesized **25** and **26** with 5-Cl or 6-Cl substituents on the benzothiazol-2-yl and found that while **25** with 5-Cl substitution is 17 times less potent than **24**, **26** with 6-Cl substitution is 4 times more potent than **24**. Our modeling further suggested that the sixth position in the benzothiazol-2-yl group can tolerate a group larger than Cl, and we replaced the 6-Cl with an isopropyl group to obtain **27**, which has a $K_i$ value of 5 nM to DCN1. Hence, **27** has a very-high affinity to DCN1 and is >10,000 times more potent than **9**.

Although **27** binds potently to DCN1, it has poor aqueous solubility (Supplementary Fig. 1), making it not suitable for cell-based studies. Turning our attention to the aqueous solubility issue, we reasoned that this poor solubility stems from the three hydrophobic groups and five peptide bonds in the molecule

(Fig. 1e). Deletion of Leu4 from **27** yielded **28**, which is 37 times less potent than **27**. Further deletion of Lys3 from **28** afforded **29**, which with a $K_i$ value of 292 nM is only slightly less potent than **28**. Conversion of the C-terminal amide in **29** into a free amino group yielded **30**, which has a $K_i$ value of 66.4 nM and is four times more potent than **29** but more significantly has excellent aqueous solubility (>20 mM at pH = 2.0 and 7.4, Supplementary Fig. 1). Since compound **30** has a molecular weight of 404 and contains only two amide bonds, it is an attractive compound for further optimization.

We next optimized the acetyl group in **30** (Fig. 1f). The co-crystal structure of UBC12 with DCN1 showed that this *N*-terminal acetyl group forms a hydrogen bond with DCN1 through its carbonyl group and its methyl group has hydrophobic contacts with a sub-pocket formed by L103, L184, T181, A106, and V102 of DCN1, suggesting the importance of this acetyl group for binding. To support this, we generated compound **31** by removal of the methyl group and demonstrated that **31** is 34-times weaker than **30** in binding to DCN1. Elimination of the entire acetyl group generated **32** that is >1000 times less potent than **30**. Our binding site analysis indicated that the sub-pocket occupied by the methyl portion of the acetyl group in **30** could accommodate a somewhat larger hydrophobic group (Fig. 1a, b) and indeed, replacement of the methyl group in **30** with an ethyl

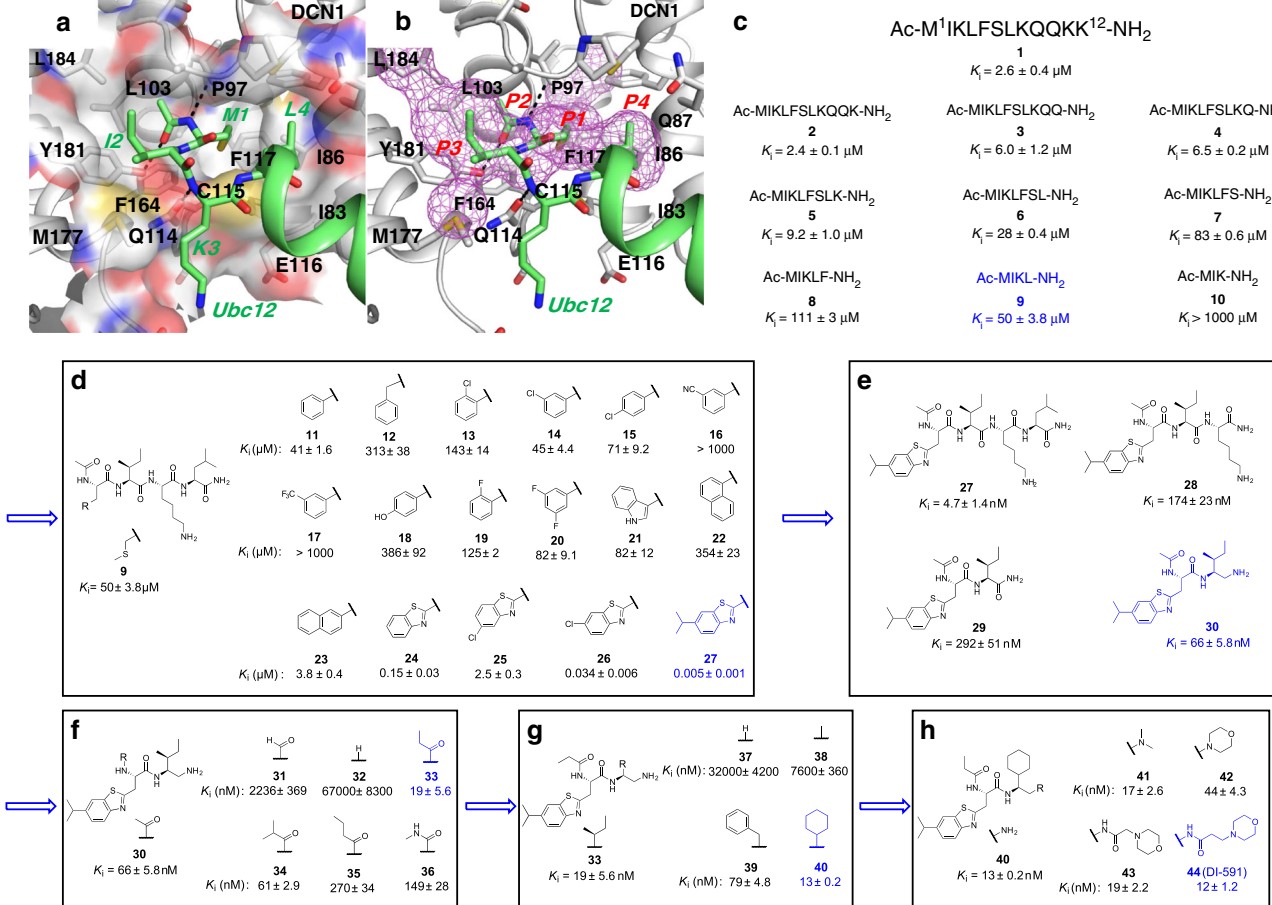

**Fig. 1** Structure-based design of inhibitors of the DCN1–UBC12 interaction. **a** Crystal structure of DCN1 complexed with UBC12 peptide (green). **b** Hydrophobic hotspots (purple mesh) at the UBC12 peptide binding site identified by co-solvent mapping. Subsites for design and optimization are labeled. **c** Truncation study of the UBC12 peptide. **d** Structures and binding affinities of tetrapeptide **9** and its analogs with Met1 modifications. **e** Structures and binding affinities of compound **27** and its analogs with reduced molecular weights and improved aqueous solubility. **f** Structures and binding affinities of compound **30** and its analogs with acetyl group modifications. **g** Structures and binding affinities of compound **33** and its analogs with Ile modifications. **h** Further optimization of the potency and physical properties of compound **40** yielding compound **44** (DI-591)

group generated **33** which, with a $K_i$ value of 19 nM, is three times more potent than **30**. Compound **34** with an isopropyl group has a $K_i$ value of 61 nM, and is equipotent to **30** but less potent than **33**. Compound **35** with an *n*-propyl group replacing the methyl group is five times less potent than **30**. Replacement of the ethyl group in **33** with a methylamino group yielded **36**, which is eight times less potent than **33**, indicating that not only the size but also the hydrophobic nature of the ethyl group is critical to achieving optimal interactions at this site.

We next modified the isobutyl side chain of Ile2 in **33** that occupies a surface hydrophobic pocket of DCN1 (Fig. 1g). Replacement of this side chain in **33** with a hydrogen atom or a methyl group generated **37** and **38**, respectively, which are >1000 and >400 times less potent than **33**. This result shows the importance of the hydrophobic interactions between the Ile isobutyl group in **33** and DCN1 for their binding affinity and subsequently, we focused on synthesizing analogs with relatively large hydrophobic groups at this site. Replacement of the isobutyl group in **33** with a benzyl group, for example, resulted in **39**, which has a $K_i$ value of 79 nM to DCN1 and is thus four times less potent than **33**. However, replacement of the isobutyl group with a cyclohexyl group generated **40**, which has a $K_i$ value of 13 nM to DCN1 and is slightly more potent than **33**.

Although **40** binds to DCN1 with high affinity ($K_i = 13$ nM) and has good aqueous solubility, its primary amino group could have non-specific cellular toxicity, thus complicating cellular investigations. Consequently, we modified this primary amine group in **40** (Fig. 1h). Dimethylation of the amine group yielded **41**, which has a $K_i$ value of 17 nM. Replacement of the amine group with a morpholino group afforded **42**, which binds to DCN1 with a $K_i$ value of 44 nM. Linking a morpholino group to the core structure of **40** through a methylene or an ethylene spacer and an amide bond resulted in **43** and **44** (DI-591), respectively, which bind to DCN1 with $K_i$ values of 18.5 and 12 nM, respectively. Compound **44** (DI-591) has excellent

aqueous solubility in both acidic and neutral conditions (>20 mM at pH 2.0 and 7.4, Supplementary Fig. 1).

Thus, starting from the 12-residue UBC12 peptide (**1**), we obtained a small-molecule DI-591 (**44**) with MW of 586 and a $K_i$ value of 12.4 nM to DCN1. DI-591 was designed to bind to those sub-pockets in DCN1 occupied by the first three UBC12 residues and is >100-times more potent than the 12-residue UBC12 peptide (**1**) and >4000 times more potent than the tetrapeptide (**9**). It also has excellent aqueous solubility and has provided us with an excellent chemical probe to examine the role of the DCN1–UBC12 protein–protein interaction in modulating the functions of cullins in cells.

**Testing affinity and specificity of DI-591 to DCN proteins**. In mammalian cells, there are five distinct DCN-like proteins (DCN1–5), which are able to stimulate neddylation of cullins with different efficiencies[30,31]. We synthesized a fluorescent-labeled probe (**46**) based on DI-591 (Fig. 2a). Saturation experiments showed that **46** binds to DCN1 and DCN2 with $K_d$ values of 21.9 and 11.2 nM, respectively, but at concentrations up to 10 μM, shows no appreciable binding to DCN3, DCN4, or DCN5 proteins (Fig. 2b). To test the binding specificity of DI-591, we synthesized DI-591DD (**45**) by changing the two chiral centers in DI-591 from L to D (Fig. 2a). DI-591DD binds to DCN1 with a $K_i$ value of 6.4 μM in our competitive FP binding assay and is thus >500 times less potent than DI-591 (Fig. 2c).

We determined the binding kinetics ($k_{on}$ and $k_{off}$) and $K_d$ values of DI-591 and DI-591DD using a label-free BioLayer Interferometry (BLI) assay (Fig. 2d, e). Our data show that DI-591 binds to DCN1 with a $K_d$ value of 30.6 nM and has a slow off-rate of 0.0038 s$^{-1}$. In comparison, DI-591DD has a $K_d$ value of 27.9 μM and a fast off-rate of 0.825 s$^{-1}$. The label-free BLI assay further confirms the high binding affinity and stereo-specificity of DI-591 to DCN1. Consistent with the binding data for **46**, DI-591 binds to DCN2 with a $K_i$ value of 10.4 nM in the competitive FP-

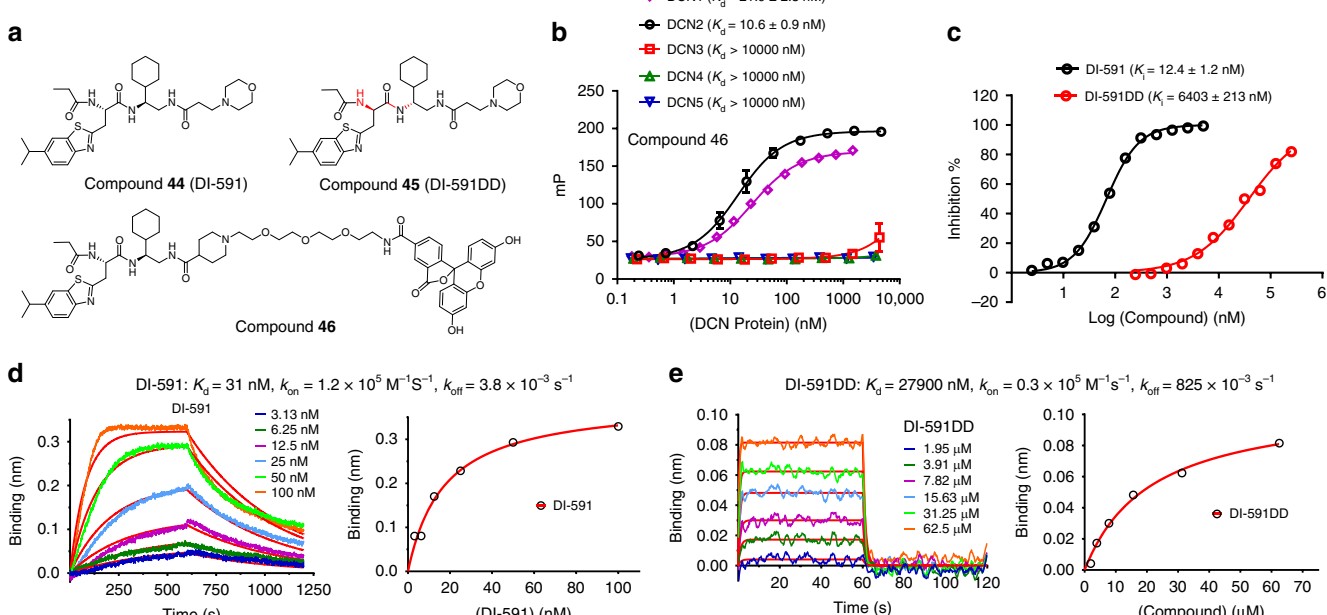

**Fig. 2** Characterization of DI-591 binding to DCN proteins. **a** Chemical structures of DI-591, DI-591DD and the fluorescent labeled compound **46**. **b** Saturation curves of **46** with indicated DCN proteins. **c** Competitive binding curves of DI-591 and DI-591DD to DCN1 using a competitive fluorescence polarization assay. **d** Kinetic binding sensorgrams of DI-591 and the steady state fitting of the equilibrium responses *vs* compound concentrations based on 1:1 binding model. **e** Kinetic binding sensorgrams of DI-591DD and the steady state fitting of the equilibrium responses vs compound concentrations based on 1:1 binding model

binding assay, and has no appreciable binding to DCN3, DCN4, and DCN5 proteins at concentrations up to 10 μM in the label-free BLI assay (Supplementary Table 1). Hence, DI-591 displays a very-high binding affinity to recombinant human DCN1 and DCN2 proteins and >1000-fold selectivity over recombinant human DCN3–5 proteins. DI-591DD binds to DCN2 with a $K_i$ value of 4.6 μM in the competitive FP-binding assay and is >400 times less potent than DI-591 (Supplementary Table 1). DI-591DD has no appreciable binding to DCN3, DCN4, and DCN5 proteins at concentrations up to 10 μM in the BLI assay (Supplementary Table 1). Because DI-591 and DI-591DD are enantiomers and have the same physicochemical properties, DI-591DD can serve as an excellent control compound in our cellular studies.

The strong binding affinities of DI-591 to DCN1 and DCN2 and its weak binding affinity to DCN3–5 proteins are consistent with the facts that the DI-591-binding site residues in DCN1 and DCN2 are identical with the exception of residue 83 (I83 vs V83) and that DCN3-5 proteins have significant differences in their binding site sequences when compared to DCN1 or DCN2 (Supplementary Fig. 2).

Since the N-terminal domain of UBC12 interacts with both DCN1 and the E1 enzyme subunit UBA3[13,17], we sought to determine whether DI-591 could directly inhibit the E1 activity, leading to blockade of neddylation of UBC12. Our data showed that while MLN4924 effectively inhibits UBC12 neddylation, DI-591 has no such effect at concentrations up to 100 μM (Supplementary Fig. 3).

**Determination of a co-crystal structure of DI-591 with DCN1.** To provide the structural basis for the high-affinity binding of DI-591 to DCN1, we determined their co-crystal structure at a resolution of 2.58 Å (Fig. 3, PDB ID: 5UFI). The detailed data

collection and refinement statistics are provided in Supplementary Table 2.

Comparison of the co-crystal structure of DI-591 complexed with DCN1 with the previously reported co-crystal structure of UBC12 complexed with DCN1 showed that while the majority of the amino acid residues of DCN1 interacting with the UBC12 peptide and DI-591 adopt very similar conformations, the F109 and F117 residues adopt significantly different conformations in these two structures (Fig. 3c). The bicyclic aromatic group of DI-591 is buried in a deep hydrophobic pocket defined by I86, F89, C90, F109, F117, A106, A111, and F164, and this deep penetration by the bicyclic ring in DI-519 induces conformational changes in F109 and F117 of DCN1 in order to provide more room for effective interactions with DI-591 (Fig. 3c). Compared to the Met1 side chain in the UBC12 protein, the bicyclic ring of DI-591 makes more extensive hydrophobic contacts with this sub-pocket in DCN1. The additional methyl group of the propionyl group in DI-591 fits nicely in the small hydrophobic pocket formed by L103, L184, Y181, A106, and V102. In addition, this propionyl group forms hydrogen bonds with P97 and Y181 of DCN1 while the amide in the soluble portion of DI-591 forms a hydrogen bond with Q114 of DCN1. The cyclohexyl group of DI-591 is half buried in a hydrophobic pocket formed by M177, A180, Y181, and L184. This co-crystal structure provides a structural basis for the high binding affinity of DI-591 with DCN1.

**Interaction of DI-591 with cellular DCN1 protein.** To determine whether DI-591 binds to cellular DCN1 and DCN2 proteins, we synthesized compound 47, a biotin-labeled analog of DI-591 (Fig. 4a). Biochemical assays show that 47 has similar high binding affinities as DI-591 to DCN1 and DCN2 recombinant proteins ($K_i = 4.0$ and 3.9 nM, respectively), and at

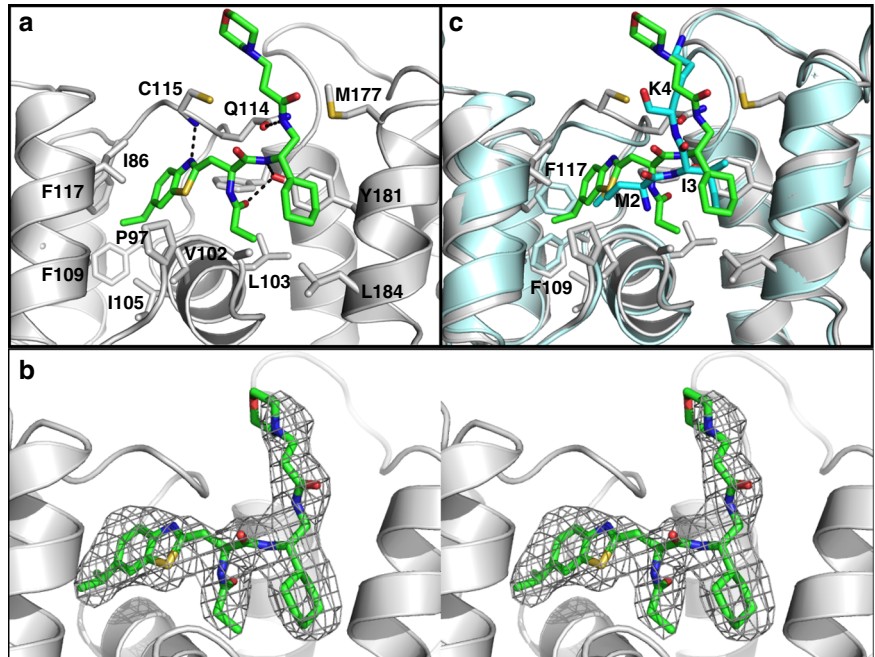

**Fig. 3** Structural basis of the high-affinity binding of DI-591 to DCN1. **a** Co-crystal structure of DCN1 (gray) complexed with **DI-591** (green). DCN1 residues interacting with DI-591 are shown as sticks with carbon atoms in gray, nitrogen atoms in blue, oxygen atoms in red and sulfur atoms in yellow. Hydrogen bonds are shown as dashed lines. **b** The stereo image of electron density map calculated prior to including **DI-591** in the structural refinement is shown as a gray grid contoured at 3σ. **c** Overlay of DCN1 (gray)–DI-591 (green) complex structure with DCN1 (light cyan)–UBC12 peptide complex structure (dark cyan) (PDB ID: 3TDU). Sidechains of residues F109 and F117 from the DCN1–UBC12 structure are shown as cyan sticks. These residues change their conformations upon DI-591 binding

concentrations up to 10 μM, has no appreciable binding to recombinant DCN3, DCN4, and DCN5 proteins.

In co-immunoprecipitation pull-down experiments, **47** efficiently pulls down both cellular DCN1 and DCN2 proteins in a dose-dependent manner in cell lysates from the KYSE70 esophageal cancer cell line (Fig. 4b) that has high DCN1 expression (Supplementary Fig. 4a). Moreover, in a competitive pull-down assay, DI-591 dose dependently inhibits the binding of **47** to both cellular DCN1 and DCN2 proteins, whereas DI-591DD has no effect (Fig. 4b). Similar pull-down results were also obtained in the KYSE140 esophageal cancer cell line with high DCN1 expression (Supplementary Fig. 4b). These results indicate that both biotinylated **47** and DI-591 potently bind to cellular DCN1 and DCN2 proteins.

We next employed the cellular thermal shift assay (CETSA)[32] to assess the target engagement of DI-591 in cells. As shown in Fig. 4c, cellular DCN1 protein was largely degraded at 55 °C in KYSE70 cells treated with DMSO or DI-591DD. The thermal stability of DCN1 protein was clearly enhanced by DI-591 at 50, and 55 °C. Furthermore, DI-591 enhances the stability of DCN1 protein in a dose-dependent manner, whereas DI-591DD has no effect (Fig. 4d). The thermal stability of DCN2 is also enhanced by DI-591 in a dose-dependent manner, although DCN2 is more sensitive to heating than DCN1 (Fig. 4d, e). These data suggest that DI-591 engages both DCN1 and DCN2 proteins in cells. Consistently, co-immunoprecipitation (Co-IP) experiments showed that the association of cellular DCN1 and UBC12 proteins was effectively reduced by DI-591 but not by DI-591DD (Fig. 4f).

Taken together, these data show that DI-591 binds to both cellular DCN1 and DCN2 proteins and disrupts the association of cellular DCN1 and UBC12 proteins.

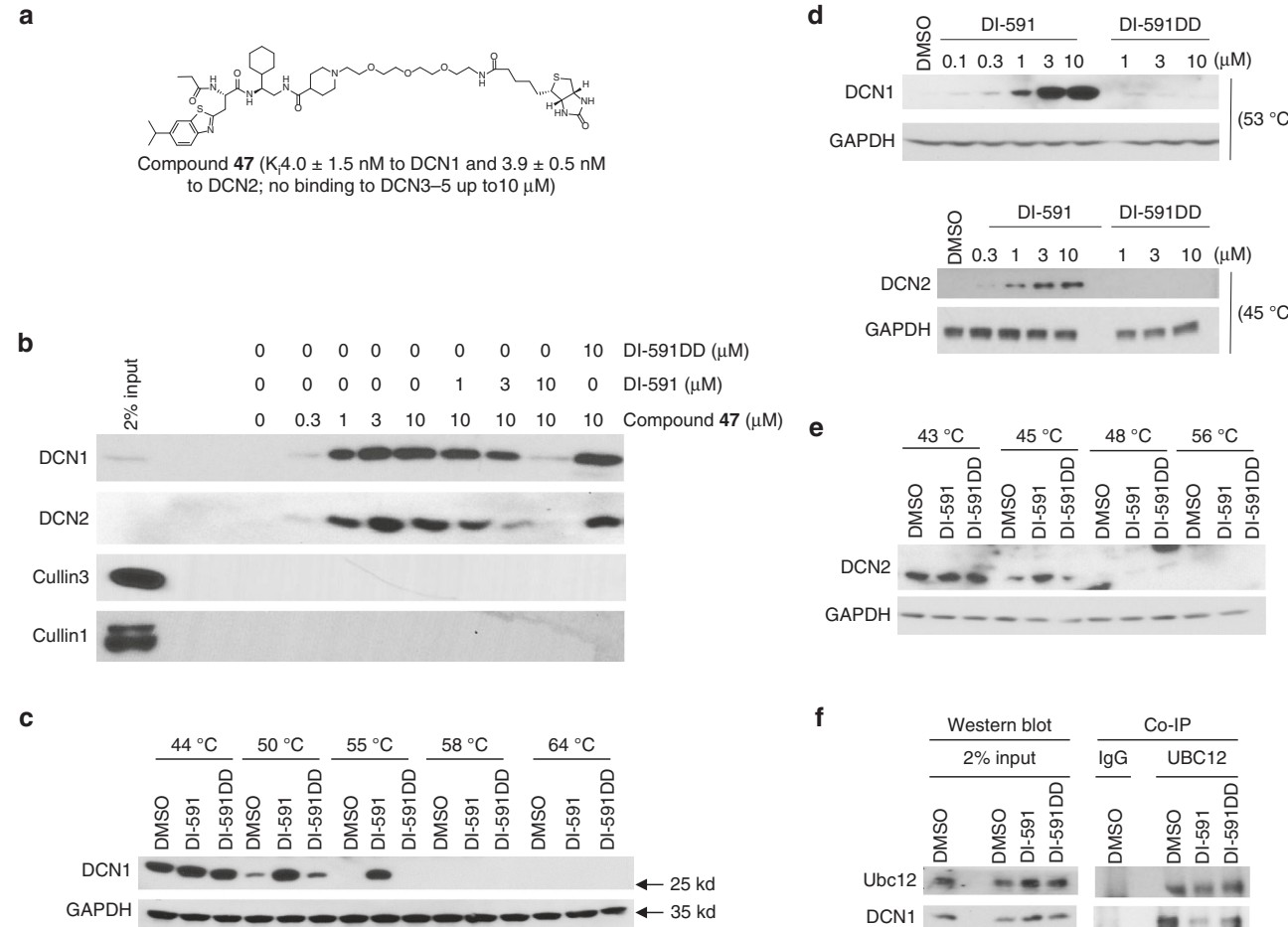

**Fig. 4** Cellular engagement of DCN proteins by DI-591. **a** Chemical structure of biotinylated compound **47** and its binding affinities to DCN1-5 proteins. **b** Pull-down of DCN1 and DCN2 protein by compound **47** and competition by **DI-591** in KYSE70 cell lysates. Protein levels of DCN1, DCN2, cullin 1 and cullin 3 pulled down from KYSE70 cell lysates with **47** alone or in combination with DI-591 or DI-591DD were examined by western blotting analysis and specific antibodies. **c** Enhancement of thermal stability of DCN1 protein by DI-591 but not by DI-591DD in KYSE70 cells. Protein levels of DCN1 in KYSE70 cells treated with DI-591 at 10 μM, DI-591DD at 10 μM or DMSO for 1 h, followed by heating at different temperatures for 3 min were examined by western blotting analysis. GAPDH was used as a loading control. **d** Enhancement of thermal stability of DCN1 and DCN2 proteins by DI-591 but not by DI-591DD treatment in KYSE70 cells. Protein levels of DCN1 and DCN2 in KYSE70 cells treated with DI-591 and DI-591DD at the indicated concentrations for 1 h and then heated at 53 °C (DCN1) and 45 °C (DCN2) for 3 min were analyzed by western blot. GAPDH was used as a loading control. **e** Enhancement of thermal stability of DCN2 by DI-591 but not DI-591DD in KYSE70 cells. Protein levels of DCN2 in KYSE70 cells treated with DI-591 at 10 μM or DI-591DD at 10 μM or DMSO for 1 h, followed by heating at different temperature for 3 min were examined by western blotting analysis. GAPDH was used as a loading control. **f** Blockage of the association of DCN1 and UBC12 in cells by DI-591 but not by DI-591DD. KYSE70 cells were treated with DI-591 at 10 μM or DI-591DD at 10 μM or DMSO for 1 h. The basal protein levels of DCN1 and UBC12 in the cell lysates were examined by western blotting analysis. Protein levels of DCN1 and UBC12 in the cell lysates pulled down by an UBC12 antibody were examined by western blotting analysis

**Selective inhibition of cullin 3 neddylation by DI-591**. Previous studies have demonstrated that DCN1 acts together with RBX1 as a co-E3 ligase to facilitate neddylation of cullin 1[15–18,33] as well as other cullins[30]. We next investigated whether DI-591 can effectively inhibit neddylation of cullin family proteins, compared to the NAE1 inhibitor MLN4924 as the control. Consistent with the previous report[21], MLN4924 at concentrations of 10–300 nM effectively suppresses neddylation of all cullins examined in a panel of six cell lines from different tissue types (Fig. 5a and Supplementary Figs. 5 and 6). While DI-591 effectively inhibits neddylation of cullin 3 in a dose-dependent manner in each of

these 6 cell lines, it surprisingly has no or at best only a modest inhibitory effect on neddylation of cullin 1 and other cullins in all of the cell lines tested (Fig. 5a and Supplementary Figs. 5 and 6). In these cell lines, DI-591 effectively inhibits neddylation of cullin 3 at concentrations as low as 0.3 μM, similar to the concentrations at which it engages cellular DCN1 and DCN2 proteins (Fig. 4b, d). Moreover, DI-591DD fails to inhibit neddylation of cullin 3 and other cullins, indicating that the inhibition by DI-591 is specific. These results clearly show that DI-591 selectively inhibits neddylation of cullin 3 but not neddylation of other cullin family members.

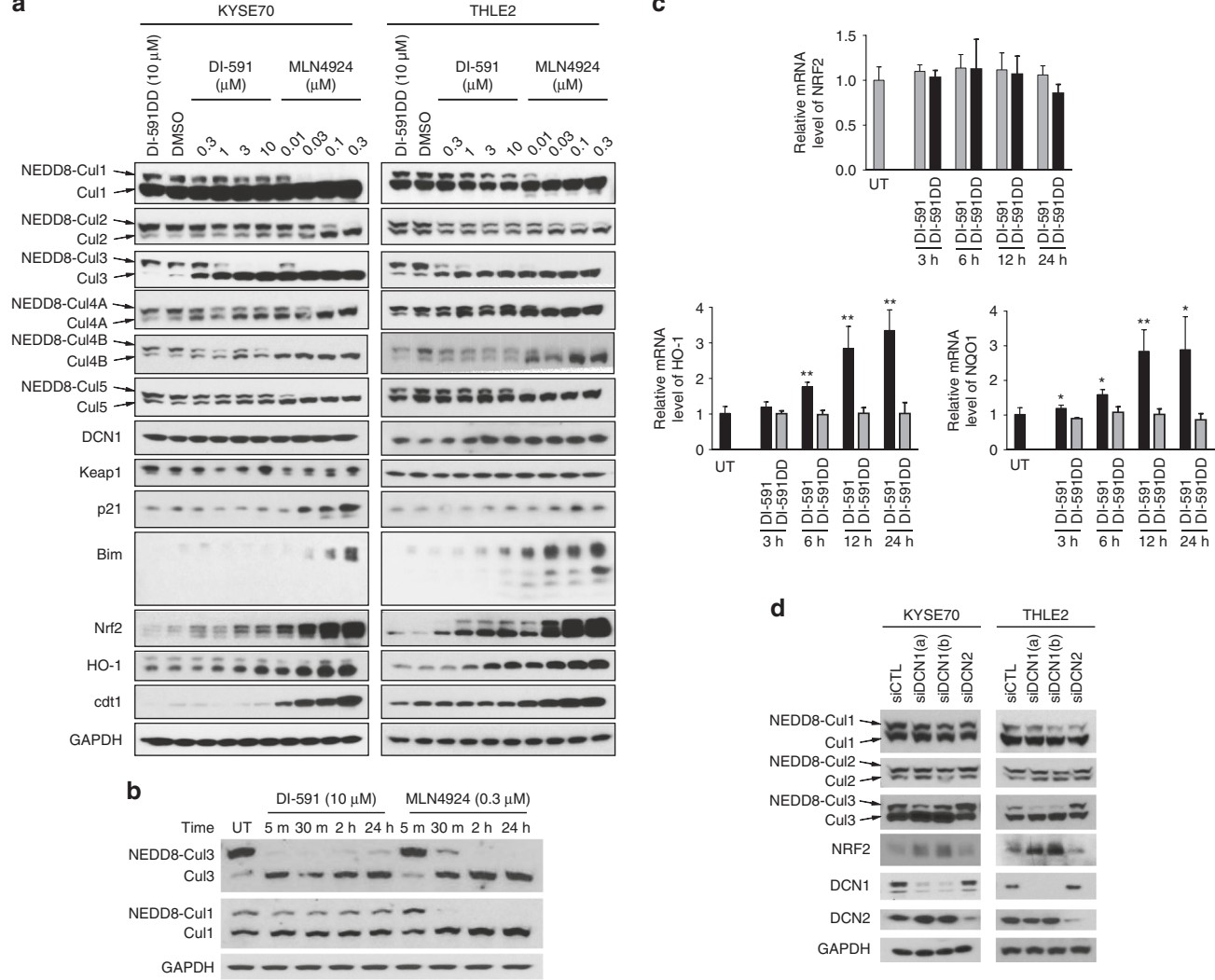

**Fig. 5** DI-591 selectively and rapidly inhibits neddylation of cullin 3. **a** Western blotting of neddylated and un-neddylated cullin family members and several representative well-known substrates of cullin CRLs in KYSE70 esophageal cancer cells and THLE2 immortalized liver cells. Cells were treated with DCN1 inhibitor DI-591 or NAE E1 inhibitor MLN4924 at indicated concentrations, or DI-591DD at 10 μM for 24 h. Protein levels of neddylated and un-neddylated cullin family members and several known substrates of cullin CRLs were examined by western blotting analysis. GAPDH was used as a loading control. **b** Inhibition kinetics of neddylation of cullin 1 and 3 by DI-591 and MLN4924 in THLE2 liver cells. Cells were treated with DI-591 at 10 μM or MLN4924 at 0.3 μM at indicated time points. Protein levels of neddylated and un-neddylated cullin 1 and 3 were examined by western blotting analysis. GAPDH was used as a loading control. **c** qRT-PCR analysis of mRNA levels of NRF2 and NRF2-reguated genes in THLE2 cells. Cells were treated with DI-591 at 10 μM, DI-591DD at 10 μM or DMSO for indicated time points. The relative mRNA levels of NRF2, NQO1 and HO-1 were examined by quantitative real-time RT-PCR assay. GAPDH was used as an internal control. The averages and standard deviations for each column were calculated from total nine samples obtained from three independent experiments (each experiment with triplicates). *P*-value from t-test: *$P < 0.05$, **$P < 0.01$. **d** Effect of siRNAs-mediated DCN1 knockdown on neddylation of cullin 1, 2, and 3. KYSE70 and THLE2 cells were transfected with small interfering RNAs against DCN1 using two different constructs or DCN2 or non-targeting siRNA oligo control (siCTL) for 48 h. The effectiveness of siRNA knockdown on DCN1 or DCN2 and the effect on the levels of neddylated- and un-neddylated cullin 1, 2, 3, and NRF2 proteins were examined by western blotting analysis. GAPDH was used as a loading control

We further examined the kinetics and selectivity of DI-591 on neddylation of cullin 3 over cullin 1 in THLE2 liver cells with MLN4924 included as the control (Fig. 5b). Remarkably, DI-591 completely depletes neddylated cullin 3 within 5 min of treatment and the effect persists throughout the treatment time, with a concurrent increase in the level of un-neddylated cullin 3 protein. DI-591 at 10 µM has no effect on neddylation of cullin 1 throughout the entire 24 h treatment period, demonstrating a striking selectivity. MLN4924 is effective in inhibition of neddylation of both cullin 1 and 3, albeit with slightly slower kinetics in inhibition of neddylation of cullin 3 than DI-591.

A key consequence of cullin neddylation is control of the E3 ubiquitin ligase activity of the cullin CRL complex, which in turn regulates protein turnover in cells. Therefore, by broadly inhibiting neddylation of all cullins[21], MLN4924 induces accumulation of substrates examined for all three cullin family members (Fig. 5a and Supplementary Fig. 5). In contrast, DI-591 dose dependently increases the protein level of nuclear factor erythroid 2-related factor 2 (NRF2), a well-known substrate for both cullin 1[34–36] and 3[24–26] CRLs, but has no or minimal effect on p21 and Bim, two substrates for cullin 1 CRL, and on CDT1, a substrate for cullin 4 A CRL[34] (Fig. 5a and Supplementary Figs. 5 and 7). Since DI-591 has no or minimal effect on p21 and Bim, two substrates for cullin 1 CRL, we thus propose that the upregulation of NRF2 is a consequence of inactivation of cullin 3 CRL by DI-591.

NRF2 is a transcriptional factor and a master regulator of antioxidant responses, regulating numerous detoxifying and antioxidant genes, such as the phase II detoxification enzymes heme oxygenase (HO1) and NAD(P)H:quinone oxidoreductase-1 (NQO1)[35]. To investigate if upregulation of NFR2 protein leads to its transcriptional activation, we examined the mRNA levels of NQO1 and HO1 in the THLE2 liver cells by qRT-PCR. DI-591, but not DI-591DD, robustly increases the mRNA levels of NQO1 and HO1 (Fig. 5c), leading to upregulation of HO1 protein in the cells (Fig. 5a). Significantly, DI-591 and DI-591DD have no effect on the mRNA level of NRF2 (Fig. 5c), indicating that the increase of NRF2 protein by DI-591 is not due to the adaptive response of cells to oxidative stress[36].

We knocked down DCN1 or DCN2 by small interfering RNA (siRNA) to directly investigate their role in neddylation of cullins. Knockdown of DCN1 by siRNA in the KYSE70 and THLE2 cell lines (Fig. 5d) results in reduced neddylated cullin 3 and accumulation of un-neddylated cullin 3, but has a minimal effect on the level of neddylated cullin 1 and cullin 2. Knockdown of DCN1 also leads to accumulation of NRF2 (Fig. 5d). Moreover, the level of accumulated un-neddylated cullin 3 and NRF2 proteins correlated with the efficiency of DCN1 knockdown by siRNA (Fig. 5d). Surprisingly, knockdown of DCN2 using siRNA has no effect on neddylation of cullin 3 and NRF2 protein levels in these two cell lines (Fig. 5d). These results indicate that DCN1, but not DCN2, plays a key role in neddylation of cullin 3 in these cell lines.

DI-591 shows no cytotoxicity to THLE2 cells and five additional cancer cell lines at concentrations up to 20 µM (Supplementary Fig. 8). The lack of cytotoxicity of DI-591 in cells suggests that selective inhibition of neddylation of cullin 3 has no negative effect on cell viability. In contrast, MLN4294, a pan-inhibitor of neddylation of all cullins, potently inhibits cell viability, with IC$_{50}$ values of 50–350 nM in these 6 different cell lines (Supplementary Fig. 8).

Collectively, these data show that by binding to DCN1 and blocking the DCN1–UBC12 protein–protein interaction, DI-591 selectively inhibits neddylation of cullin 3 but has no or minimal effect on neddylation of other cullin family members. This selective inhibition of neddylation of cullin 3 by DI-591 leads to

accumulation NRF2 protein and its transcriptional activation. Knockdown experiments indicate that DCN1, but not DCN2, plays a key role in regulation of neddylation of cullin 3 but not of other cullins.

## Discussion

Because the CRLs regulate homeostasis of ~20% of cellular proteins and their activation requires neddylation of their respective cullin subunit by NEDD8, targeting neddylation of individual cullin members may be a very attractive strategy for selective control of the levels of certain cellular proteins. MLN4924 was designed to bind covalently to NEDD8, thus achieving non-selective, efficient and broad inhibition of neddylation of all cullin family members. To date, selective inhibition of neddylation of individual cullin members by small-molecule inhibitors has not been achieved.

DCN1 functions as a co-E3 ligase in the neddylation pathway, but its role in the regulation of specific cullin members in cells is unclear. Based upon the biochemical and structural data, we hypothesized that the DCN1–UBC12 protein-protein interaction is amenable to pharmacological blocking by small-molecule inhibitors. We further postulated that pharmacological blocking of the DCN1–UBC12 protein-protein interaction might be effective in inhibition of neddylation of cullin 1 because DCN1 is an integral part of the cullin 1 complex. Since no potent and cell-permeable small-molecule inhibitor of the DCN1–UBC12 protein–protein interaction was available[37], we first undertook the task of creating such a compound, starting from the native 12-mer UBC12 peptide and performing structure-based design and extensive medicinal chemistry optimization. Our efforts have resulted in the discovery of DI-591 as a potent and cell-permeable small-molecule inhibitor that binds to DCN1 with a high affinity and blocks the DCN1–UBC12 protein–protein interaction in cells.

Contrary to our initial hypothesis, our data showed that DI-591 at best only has a very modest effect on inhibition of neddylation of cullin 1 in cells. Rather, it is very effective and potent in the inhibition of neddylation of cullin 3 in all cell lines examined but is ineffective in the inhibition of all other cullin family members that we have examined. Therefore, our data firmly establish that DI-591 is a potent and selective inhibitor of neddylation of cullin 3 over other cullin members in cell lines of diverse tissue types. This effect is quite specific since DI-591DD, an enantiomeric control compound, fails to show any significant inhibition of neddylation of cullin 3 or any other cullin members. Our knockdown experiments using siRNA further indicate that while DCN1 is required for neddylation of cullin 3, it is not required for neddylation of cullin 1.

Although DI-591 effectively interacts with both cellular DCN1 and DCN2 proteins (Fig. 4 and Supplementary Fig. 4b), knock-down of DCN1 leads to effective and selective inhibition of neddylation of cullin 3, whereas surprisingly, efficient knockdown of DCN2 has no effect on neddylation of cullin 3 (Fig. 5d). Furthermore, using a validated DCN2 antibody, we failed to detect the expected full-length DCN2 protein in all the cell lines examined. Instead, the detected DCN2 protein in our study has a MW of 21 kDa, which matches the theoretical MW of the shorter DCN2 splicing isoform (sequence: 1–186)[38]. We analyzed the Cul1-RBX1-UBC12-NEDD8-DCN1 complex structure[18] and found that while the detected DCN2 isoform retains the UBC12 binding site, it lacks the C terminus domain, which is required for interaction with the cullin protein. We therefore propose that the short splicing isoform of DCN2 detected in all the cell lines in our study is not functionally equivalent to the DCN1 protein and the cellular target for the effective inhibition of neddylation of cullin 3

by DI-591 is DCN1, but not DCN2. However, DI-591 should inhibit the function of full-length DCN2 protein in cells where this isoform is expressed.

Although cells have three other DCN proteins (DCN3-5), DI-591 fails to bind to them. Hence, although DCN3-5 proteins also bind to UBC12, our data (Fig. 5a and Supplementary Figs. 5 and 6) suggest that once the interaction of DCN1 with UCB12 is blocked by DI-591, DCN3-5 cannot substitute DCN1 to maintain the neddylation activity for the cullin 3 complex.

As previously shown in in vitro assays by Monda et al.[30], DCN1 promotes UBC12-associated neddylation of cullin 1, 2, 3, and 4. However, DCN1 binds to cullin 3 more potently than to other cullin family members. This affinity difference may partially account for our observed remarkable selectivity for DI-591 in inhibition of neddylation of cullin 3 over other cullins. However, the underlying precise mechanism for this remarkable selective inhibition of neddylation of cullin 3 by DI-591 over other cullin members clearly deserves further investigation.

We examined the cellular consequences of selective inhibition of neddylation of cullin 3 by DI-591 (Fig. 5a and Supplementary Figs. 5 and 7). DI-591 is quite effective in inducing accumulation of NRF2 protein and the transcriptional activation of NRF2, leading to upregulation of HO1 and NQO1 at both mRNA and protein levels. The induced accumulation of NRF2 protein by DI-591 is not due to oxidative cellular stress because DI-591 does not induce upregulation of NRF2 mRNA in cells.

Of note, NRF2 is a substrate for both cullin 1 and 3 CRLs[24–26,39–41]. Since DI-591 fails to upregulate p21 and Bim, two well-recognized substrates for cullin 1 CRL (Fig. 5a and Supplementary Fig. 5), we conclude that the upregulation of NRF2 by DI-591 is due to its selective inhibition of cullin 3 CRL. This conclusion is further supported by our data that MLN4924 induces much stronger upregulation of NRF2 than DI-591 in all

cell lines examined (Fig. 5a and Supplementary Fig. 5), due to effective inhibition of both cullin 1 and 3 CRLs by MLN4924 and selective inhibition of cullin 3 CRL by DI-591.

Consistent with its broad inhibition of neddylation of all cullin members, MLN4924 is highly cytotoxic against all the cell lines we examined. In sharp contrast, DI-591 shows no significant cytotoxicity at concentrations as high as 20 μM, highlighting a major difference between the broad inhibition of neddylation of all cullin family members and the selective inhibition of cullin 3 neddylation.

Dimethyl fumarate activates NRF2 by covalently binding to KEAP1 and has been approved for the treatment of relapsing multiple sclerosis[42]. The ability of DI-591 to induce robust upregulation of NRF2, coupled with its lack of cytotoxicity, suggests that DI-591 and its analogs should be evaluated as potential therapeutic agents for the treatment of multiple sclerosis and other human diseases in which strong upregulation of NRF2 may be beneficial.

We propose a model (Fig. 6) to summarize the mechanism of action of DI-591 for its effective inhibition of neddylation of cullin 3. By binding to both UBC12 and cullin 3, DCN1 promotes the formation of a functional complex consisting of RBX1-UBC12~NEDD8-Cul3-DCN1 for efficiently transferring NEDD8 from UBC12 to the K712 residue of cullin 3, leading to activation of cullin 3 CRL ubiquitin ligase function and subsequent degradation of substrate proteins. Blocking the DCN1–UBC12 interaction by DI-591 results in inactivation of this functional complex, which disables neddylation of cullin 3 and leads to accumulation of substrate proteins, such as NRF2.

In summary, the present study describes the structure-based discovery of DI-591 as a potent, specific and cell-permeable small-molecule inhibitor of the DCN1–UBC12 protein–protein interaction. Using DI-591, we have made the surprising discovery

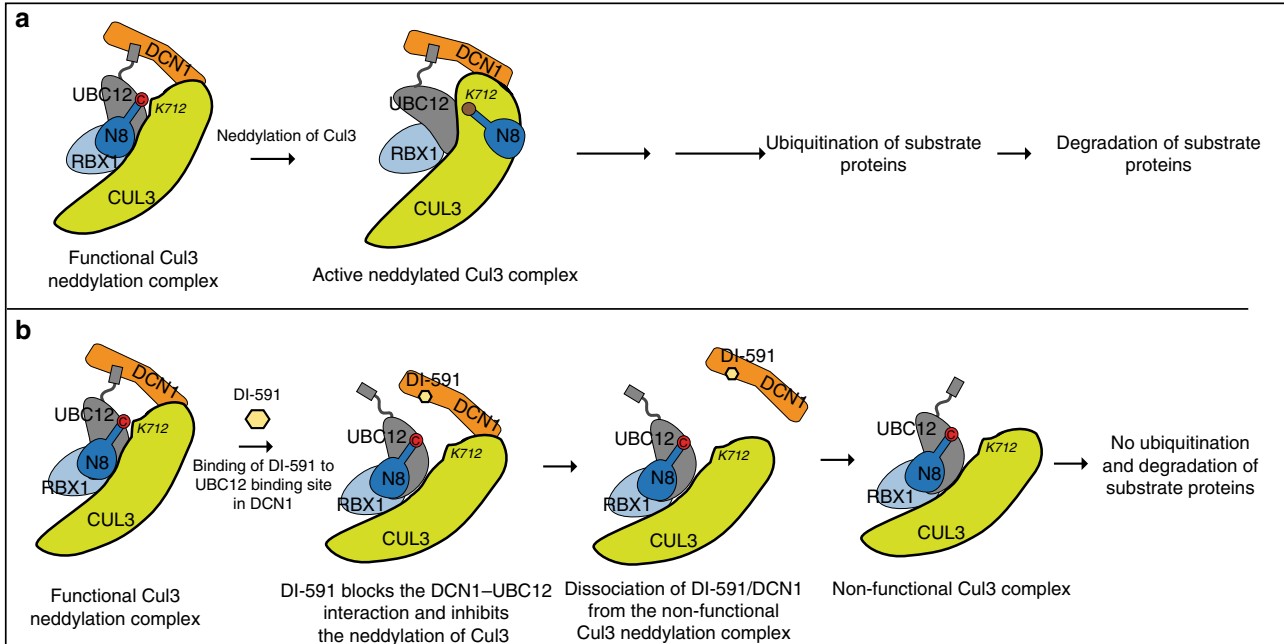

**Fig. 6** Proposed model for inhibition of cullin 3 neddylation by DI-591. **a** Proposed activation of the cullin 3 neddylation complex. DCN1, functioning as a co-E3, is required for the formation of a functional cullin 3 neddylation complex to facilitate the transfer of NEDD8 from UBC12 to the K712 residue of cullin 3. The active neddylated cullin 3 complex proceeds to form a functional cullin 3 ring ubiquitin ligase complex that performs ubiquitination of substrate proteins, ultimately resulting in substrate protein degradation by the proteasome. **b** Proposed model for inactivation of the functional cullin 3 complex by DCN1 inhibitor DI-591. DI-591 binds to the UBC12 binding site in DNC1, leading to dissociation of DCN1 from the functional complex and resulting in an inactive cullin 3 neddylation complex

that blocking the DCN1–UBC12 interaction leads to selective inhibition of neddylation of cullin 3 over neddylation of other cullin family members. DI-591 and its enantiomeric control are valuable probe compounds to further investigate the role of cullin 3 CRL in biological processes and human diseases.

## Methods

**Chemistry**. The synthesis schemes, procedures and characterization of the compounds are provided in the Supplementary Methods. The $^1H$ and $^{13}C$ NMR spectra of the compounds in this article are included in Supplementary Figs. 23–61.

**Computational methods**. DCN1 protein structure was extracted from the reported complex structure (PDB ID 3TDU) and utilized for computational co-solvent simulations in Amber with the ff99SB force field[43]. 16 ns simulations were performed in 20% v/v isopropyl alcohol/water co-solvent media following a protocol described previously[27–29]. Initial alignment of saved conformations was performed using the ptraj utility from the Amber suite and hotspots were determined based on grid occupancy of co-solvent atoms.

**Competitive FP binding assay**. The FP competitive binding experiments were performed to determine the binding affinities of all synthesized DCN1 inhibitors. A FAM labeled fluorescent probe compound (**46**) was designed and synthesized based on one of our potent small-molecule DCN1 inhibitors. Equilibrium dissociation constant ($K_d$) values of **46** to DCN1 and DCN2 proteins were determined from protein saturation experiments by monitoring the total FP values of mixtures consisting of the fluorescent probe at a fixed concentration and either DCN1 or DCN2 protein with increasing concentrations up to full saturation. Serial dilutions of DCN1 or DCN2 protein were mixed with the probe (**46**) to a final volume of 200 μl in the assay buffer (100 mM phosphate buffer, pH = 6.5, with 0.02% Tween-20 and 2% DMSO). The final probe concentration was 5 nM in both DCN1 and DCN2 assays. Plates were incubated at room temperature for 30 min with gentle shaking for equilibrium. FP values in millipolarization units (mP) were measured using the Infinite M-1000 plate reader (Tecan US, Research Triangle Park, NC) in Microfluor 1 96-well, black, round-bottom plates (ThermoFisher Scientific, Waltham, MA) at an excitation wavelength of 485 nm and an emission wavelength of 530 nm. $K_d$ values of **46** were calculated by fitting the sigmoidal dose-dependent FP increases as a function of protein concentrations using Graphpad Prism 6.0 software (Graphpad Software, San Diego, CA).

The $IC_{50}$ and $K_i$ values of compounds were determined in competitive binding experiments. Mixtures of compounds in DMSO (4 μl) and of pre-incubated protein/probe complex solution (196 μl) in the assay buffer were added into assay plates that were incubated at room temperature for 30 min with gentle shaking. Final concentrations of DCN1 protein and fluorescent probe (**46**) were 50 and 5 nM, respectively. Negative controls containing protein/probe complex only (equivalent to 0% inhibition), and positive controls containing only free probes (equivalent to 100% inhibition), were included in each assay plate. FP values were measured as described above. $IC_{50}$ values were determined by nonlinear regression fitting of the competition curves using the dose response inhibition equation (four parameters, variable slopes) included in the Prism/GraphPad software. $K_i$ values were calculated using the methods described earlier[44].

**Biolayer interferometry method**. Biolayer interferometry (BLI) kinetic experiments were performed using an OctetRED96 instrument from PALL/ForteBio. All assays were run at 30 °C using PBS (pH 7.4) as the assay buffer, in which 0.1% BSA and 0.01% Tween-20 were added to reduce non-specific interactions. 2% DMSO was introduced to increase compound solubility. Assays were run in Greiner 96-well black flat-bottom microplates in which protein solutions, pure assay buffer for dissociation, and serial dilutions of a tested compound were loaded. During the entire experimental progress, sample plates were continuously shaken at 1000 RPM to minimize any mass transport effect.

Biotinylated DCN1 protein prepared using the Thermo EZ-Link long-chain biotinylation reagent was tethered on Super Streptavidin (SSA) biosensors by dipping sensors into plate wells with protein solutions whose concentrations were pre-determined from preliminary control experiments to achieve the best signal to noise ratio. Sensor saturation typically was achieved in 10–15 min. Biotinylated blocked Streptavidin (SAV-B4) sensors were also prepared by following the protocol provided by the manufacturer as the inactive reference controls.

Sensors tethered with proteins were moved and dipped into wells with pure assay buffer and equilibrated in the buffer for 10 min to eliminate loose non-specific bound protein and establish a stable base line. Association–dissociation cycles of compounds were performed by moving and dipping sensors into compound solution wells then into pure buffer wells. Association and dissociation times were carefully determined to assure complete association and dissociation.

DMSO-only reference was included in all assays. Raw kinetic data collected were processed with the Data Analysis software provided by the manufacturer using double reference subtraction in which both DMSO-only reference and inactive protein reference were subtracted. The resulting data were analyzed based

on 1 : 1 binding model from which $k_{on}$ and $k_{off}$ values were obtained and $K_d$ values were then calculated.

**Aqueous solubility assay**. The equilibrium aqueous solubility of an individual compound at different pH values was determined with a small-scale shake-flask method in Eppendorf tubes. Compounds were added as powder in excess to Britton-Robinson buffer at pH 2.0 or phosphate buffer at pH 7.4 until heterogeneous suspensions were obtained. Suspensions containing excessive compound powder were sonicated in a water bath for 30 min followed by incubation at room temperature with shaking for 24 h to achieve thermodynamic equilibrium. Suspensions were centrifuged at 15,000 r.p.m. for 15 min. Supernatants obtained were filtered using Millipore MultiScreen filter plates equipped with 0.45 μm membranes. Compound concentrations in the saturated solutions were measured by UV spectroscopy at the maximum absorption wavelength of each compound that were pre-determined by a spectrum scan.

**In vitro UBC12-NEDD8 thioester forming assay**. NEDD8, NEDD8 E1 (APP-BP1/Uba3) and UBC12 proteins were purchased from Boston Biochem. NEDD8 (10 μM), NEDD8 E1 (25 nM), and UBC12 (2 μM) were incubated in Tris-HCl (pH 7.4, 50 mM), $MgCl_2$ (5 mM), DTT (0.5 mM) and BSA (0.1 mg/ml) with a tested compound at 25 °C for 10 min. Reactions were initiated by addition of 2 mM ATP (or distilled $H_2O$) and ran for 5 min at 25 °C. 1 M DTT (or distilled $H_2O$) was added to the reaction mixtures which were then kept on ice for 1 min followed by addition of SDS loading buffer (without DTT). The products were separated by SDS–PAGE, visualized by Coomassie stain and quantified by ImageLab (Bio-Rad).

**Cloning and purification of DCN proteins**. Human DCN1 (residues 58–259) were cloned into a pDEST17 plasmid containing an N-terminal His$_6$ tag. DCN2 (residues 62–259), DCN3 (residues 86–304), DCN4 (residues 102–292) and DCN5 (residues 47–237) were cloned into an N-terminal His$_6$-TEV expression vector. Pure proteins were derived from the same expression and purification protocols. Plasmids were transformed into Rosetta2 cells. The cells were grown in Terrific Broth at 37 °C until OD$_{600}$ > 1.0 then induced with 0.4 mM isopropyl β-D-1-thiogalactopyranoside overnight at 20 °C. The pelleted cells were resuspended in lysis buffer containing Tris-HCl (25 mM, pH 7.50), NaCl (200 mM) and protease inhibitors, then sonicated and centrifuged at 34,000 × g for 45 min to remove debris. Cleared lysate was incubated with Ni-NTA resin (Qiagen) prewashed with lysis buffer, for 1 h at 4 °C. The matrix was loaded into a column then washed with Tris-HCl, pH 7.5 (25 mM), NaCl (200 mM) and imidazole (10 mM). Protein was eluted with Tris-HCl, pH 7.5 (25 mM), NaCl (200 mM) and imidazole (300 mM), then concentrated and applied to a Superdex 75 (GE Healthcare) column pre-equilibrated with Tris pH 7.5 (25 mM), NaCl (200 mM) and DTT (1 mM). For DCN2-5, the N-terminal His$_6$ tag was removed prior to gel filtration. Tag removal was achieved through incubation with TEV protease during overnight dialysis against Tris pH 7.5 (25 mM), NaCl (200 mM) and DTT (1 mM) and a second Ni-NTA column. DCN2-5 proteins were stored at −80 °C in 1 mg/mL fractions containing 5% glycerol. The uncleaved DCN1 protein was stored at −80 °C without glycerol.

**X-ray structural determination of DCN1:DI-591 complex**. Prior to crystallization, DCN1 in Tris-HCl pH 7.5 (25 mM), NaCl (200 mM) and DTT (1 mM) was concentrated to 10 mg/mL and incubated for 1 h at 4 °C with DI-591 in a 1:1.3 protein to compound molar ratio. Crystals were grown at 20 °C from sitting drop vapor diffusion experiments. Drops contained DCN1:DI-591 (1 μl) and well solution (20% PEG 4000 and 100 mM monobasic potassium phosphate (1 μl).

Prior to data collection, crystals were cryoprotected with well solution containing 25% ethylene glycol. Diffraction data were collected on a Mar225 detector mounted on the LS-CAT 21-ID-F beamline at the Advanced Photon Source and processed with HKL2000[45]. The structure was solved by molecular replacement (Molrep[46]) using an in-house DCN1 structure lacking its bound ligand as the search model. The resulting DCN1:DI-591 structure had four protein molecules in the asymmetric unit, each containing one bound DI-591 molecule. The structure was iteratively fit and refined to 2.58 Å resolution using Coot[47] and Buster[48], respectively. The coordinates and restraints for the compound were determined using Grade[49] with the mogul + qm option. Residues 60–251were visible in the electron density maps. Data collection and refinement statistics are provided in Supplementary Table 2.

**Cell lines and culture conditions**. Immortalized liver THLE2 (ATCC CRL-2706), MDA-MB-231, U2OS, HepG2 and Hela cell lines were purchased from ATCC (Rockville, MD). Esophageal cancer cell lines KYSE70 and KYSE140 were purchased from DSMZ (Braunschweig, Germany). The THLE2 cell line was maintained in BEGM from Lonza/Clonetics Corporation (CC3170, Walkersville, MD) and the other cell lines were maintained in RPMI1640, supplemented with 10% FBS and pen–strep at 37 °C in a humidified incubator with 5% $CO_2$.

**Immunoblotting and antibodies**. Treated cells were lysed by RIPA buffer supplemented with protease and phosphatase inhibitors. The expression level of

indicated proteins was examined by immunoblotting analysis. GAPDH was used as the loading control. Anti-Cullin 1 (sc-11384), -Cullin 2 (sc-10781), and -Keap1 (sc-33569) antibodies were purchased from Santa Cruz Biotechnology (Santa Cruz, CA); anti-Cullin 4A (PA5-14542), -Cullin 4B (PA5-35239) and -DCN3 (DCUN1D3, PA5-44000) antibodies from ThermoFisher Scientific (Wayne, MI); anti-Cullin 3 (2759), -NRF2 (12721), -HO1 (70081), -Bim (2819), -cdt1 (3386) and -p21 (2947) antibodies from Cell Signaling Technology (Boston, MA); anti-DCN1 (GWB-E3D700) antibody from GenWay Biotech (San Diego, CA). Anti-Cullin 5 antibody (A302-173A) from Bethyl Laboratories (Montgomery, TX); anti-DCN2 antibody (DCUN1D2, ARP68256_P050) from Aviva Systems Biology (San Diego, CA); anti-UBC12 (14520-1-AP) from Proteintech (Rosemont, IL). Results are representative of three independent experiments.

**Biotinylated protein pull-down assay.** KYSE70 cells were lysed with RIPA buffer, and the whole cell lysate was incubated with biotinylated compound (47) alone or co-incubated with either DI-591 or DI-591DD for 1 h. Complexes formed between the biotinylated DI inhibitor and its targeted proteins were recovered by incubation with Streptavidin-agarose beads (Thermo Scientific Pierce, Waltham, Massachusetts). DCN1 and DCN2 proteins associated with beads were eluted by heating and detected by immunoblotting. Results are representative of three independent experiments.

**Co-immunoprecipitation assay.** KYSE70 cells treated as indicated for 1 h were lysed with RIPA buffer, and the whole cell lysate was incubated with anti-UBC12 antibody (14520-1-AP, Proteintech). Complexes associated with the antibody were recovered by incubation with Protein A/G PLUS-Agarose (sc-2003, Santa Cruz Biotechnology). DCN1 and UBC12 proteins associated with beads were eluted by heating and detected by immunoblotting. Results are representative of three independent experiments.

**Cellular thermal shift assay.** Cellular Thermal Shift Assay was performed according to the reported method[32]. Briefly, cells ($5 \times 10^5$ per sample) were treated with a compound or with DMSO for 1 h, washed with PBS three times, and dissolved in 50 μl PBS supplemented with a protease inhibitor, followed by heating at the indicated temperatures in a Mastercycler gradient (Eppendorf, New York, USA). Treated cells were then subjected to snap-freezing in liquid nitrogen and thawed on ice for 3 cycles. The protein levels of DCN1 and DCN2 in equal amounts of the supernatant were examined by western blots. GAPDH was used as the control. Results are representative of three independent experiments.

**siRNA assay.** KYSE70 and THLE2 cell lines were transfected with siRNAs against DCN1 (J-019139-05-0002, GE Dharmacon, Lafayette, CO), DCN1(a), DCN1(b) or DCN2 (L-020261-01-0005, GE Dharmacon) with Lipofectamine RNAiMAX Transfection Reagent (13778075, ThermoFisher Scientific Inc). After transfection for 48 h, cells were collected and analyzed by western blotting.

**Quantitative reverse transcription PCR assay.** Total RNA in treated cells was isolated with the RNeasy Mini Kit (Qiagen, Germantown, MD). RNA was reverse transcribed into complementary DNA (cDNA) using high capacity RNA-to-cDNA kit (Applied Biosystems, Carlsbad, California). The mRNA expression was examined by quantitative reverse transcription PCR (RT-qPCR) assay with TaqMan Gene Expression Master Mix (4370074). GAPDH was used as an internal reference. The level of NRF2, NQO1 and HO-1 gene expression was expressed as levels relative to untreated control. Real-time PCR primers (probes) for GAPDH (Hs02786624_g1), Nrf2 (Hs00975961_g1), NQO1 (Hs01045993_g1) and HO-1 (Hs01110250_m1) were purchased from ThermoFisher Scientific. Error bars correspond to SD from three independent experiments.

**Data availability.** The co-crystal structure of DI-591 in a complex with DCN1 was deposited in the Protein Data Bank under accession code 5UFI. All the data that support the findings of this study are available from the corresponding author upon request.

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

## Acknowledgements

This work was supported in part by National Key R&D Program of China (2016YFA0501800). Use of the Advanced Photon Source, an Office of Science User Facility operated for the US Department of Energy (DOE) Office of Science by Argonne National Laboratory, was supported by the US DOE under Contract No. DE-AC02-06CH11357. Use of the LS-CAT Sector 21 was supported by the Michigan Economic Development Corporation and the Michigan Technology Tri-Corridor (Grant 085P1000817). This work was also supported in part by the National Cancer Institute, National Institutes of Health (CA156744 and CA171277 to Y.S.).

## Author contributions

H.Z. designed and synthesized compounds in this work and wrote the manuscript. J.L. designed and performed cell biology experiments and wrote the manuscript. L.L. designed and performed all the biochemical and biophysical experiments and wrote the manuscript. D.B. designed and performed computational analysis and structure-based design and wrote the manuscript. J.S. designed and performed experiments related to protein expression and x-ray crystallography and wrote the manuscript. Z.P. contributed to writing and revising of the manuscript. K.C. and S.L. performed experiments related to protein expression and X-ray crystallography. Q.Y. and W.Z. performed experiments related to expression of different DCN proteins in cells and tested designed compounds. C.-Y.Y. modeled the complex structure and contributed to writing of the manuscript. E.F.-S. designed and performed certain cell biology experiments and wrote the manuscript. S.W. and Y.S. initiated the project, designed experiments, analyzed data, and wrote the manuscript.

## Additional information

**Competing interests:** The authors declare no competing financial interests.

