## [Peer Review File · Nature Communications]

Reviewers' comments:

Reviewer #1 (Remarks to the Author):

This paper reports solid work leading to the discovery of a novel chemical probe targeting the ubiquitin proteasome system. Starting from a crystal structure solved by the Schulman lab in 2014 (Scott, D. C. et al. Cell 157, 1671–1684 (2014)) of the pentameric complex RBX1-UBC12~NEDD8-CUL1CTD-DCN1, which showed that the E2 UBC12 docks its C-terminal tail into a deep groove of the DCN1 protein, the authors embarked in a systematic and well-executed medicinal chemistry campaign combining structure-guided peptidomimetic approaches yielding compound 44 (DI-591) binding DCN1 with high-affinity (K_i of 10 nM based on a competitive FP assay). The biophysical data to support bona-fide reversible interaction of the compound to its target is robust, employing BLI as well as FP, and showing that the inhibitor binds with similar potency to both Dcn1 and (presumably) paralogous protein Dcn2, but not to Dcn3-5. They also synthesized an enantiomer compound and showed that it binds with much weaker affinity (>500 fold less potent), making it an ideal negative control compound for cellular studies. The authors then solved the crystal structure of the compound bound to Dcn1 using co-crystallization. The structure confirmed the binding mode expected based on the rational design, and revealed some important differences relative to the bound peptide that the authors appropriately describe.

The team goes on to use compound 47, a biotinylated analogue of the most potent inhibitor, in target pull-down experiments from cell lysates. Here the first unexpected result is seen, i.e. that the biotinylated compound pulls down in a specific fashion Dcn1, but it does not pull down Dcn2 at all (Fig. 4b). This result is confirmed using CETSA in both temperature dependent (Fig 4c) and dose-dependent manner (Fig. 4d). co-IP experiments confirmed DI-591 blockade of the Dcn1-Ubc12 protein-protein interaction in cells (fig. 4e). Finally, functional characterization of the chemical probe in three different cell lines – THLE2 liver cell line (Fig. 5), liver cancer HepG2 cells (Supp Fig 4) and KYSE70 esophageal cancer cell line (S. Fig 3) – gave a second unexpected result: that is the inhibitor appear to only block neddylation of Cul3 but not any of the other Cullins tested (Cul1, Cul2, Cul4A/B and Cul5) in a dose-dependent manner. This was surprising given Dcn1 had been involved in Cul1 neddylation. Because blockade of Cul3 neddylation is expected to provide a more limited impact in stabilizing proteasomal substrates, compared to eg blockade of neddylation of all Cullins using MLN4924, the authors elegantly assess the impact of their Dcn1 chemical probe on protein levels and activity of Nrf2 – a known Cul3 substrate via the Cul3-Keap1 CRL activity. They show that Nrf2 protein levels go indeed up in concentration dependent manner, but not levels of non-Cul3 substrates e.g. Cyclin E (a Cul1 substrate) and others (Fig. 5a). This leads to upregulation of downstream Nrf2 target genes such as HO-1 and NQO1 at both mRNA level (Fig. 5c) and protein level, but no change in Nrf2 mRNA level, consistent with the activity being due to post-translational stabilization of Nrf2 via blockade of its ubiquitination and degradation. Finally, the authors show that the Cul3-selectivity of their Dcn1 inhibitor is to a degree consistent with siRNA knockdown of Dcn1. This is a somewhat tricky experiment that may not be as conclusive as expected, because it is known that target inhibition can have very different mechanistic consequences from target removal, which in any case is usually incomplete and requires long treatments with siRNA, leading to potential compensatory effects.

Overall, this is a nice article that discloses a novel and potentially useful chemical probe for studying the function of Dcn1, and in particular, if further validated, the selective blockade of Cul3 neddylation. It is a first in the field, and a worthy advance, and as such I believe that it would eventually merit publication in a journal of the calibre of Nature Communication. However, in my opinion the work suffers from several shortcomings, and that the evidence as presented does not support fully the claims made – as highlighted below – which warrants these issues to be suitably addressed experimentally in a revised version before the paper is ready for publication.

Major points:

1) The first unexpected result concerns the lack of inhibitor binding to cellular Dcn2. This contradicts the biophysical data in vitro using recombinant proteins, and warrants further investigation. Firstly, the Dcn2 antibody used should be shown to be siRNA or CRISPR proof, to

really validate that it is detecting the expected protein. The authors should also provide more information on (paralogs?) Dcn2-5, e.g. how they compare / differ relative to Dcn1, and reference any relevant literature. What are the sequence identities for example in the domain where the Ubc12 and inhibitors bind? A figure including bars showing the different lengths and domain arrangements of the five proteins would be helpful. Ultimately, the reader would be left wondering why does the probe not bind to native Dcn2? Can the author provide additional evidence and speculate on possible reasons?

2) The conjugation of Nedd8 to Cullins had been shown to require a ubiquitin-like enzyme cascade involving the Nedd8-activating enzyme E1, the Nedd8-conjugating enzyme E2 (Ubc12), the RING protein Rbx1, and the activator E3 Dcn1, resulting in neddylation of Cul1 at lysine 720 (Kurz et al., 2005 Nature 435, 1257–1261; Pan et al., 2004 Oncogene 23, 1985–1997). These two articles provide key discoveries related to the Dcn1 dependent neddylation pathway and should be cited. Further, Schulman et al had shown that RBX1 and Dcn1 function synergistically via a “dual E3” mechanism (Scott, D. C. et al. 2010 Mol. Cell 39, 784–796.), mainly on Cul1 as previously shown, but also on other Cullins. It is therefore quite surprising that the Dcn1 inhibitor appears to affect neddylation of only Cul3 but not the other cullins.

Firstly, the authors are encouraged to provide additional evidence to validate and support this remarkable discovery. Is the Cul3 antibody used siRNA or Crispr proof?

This reviewer found it curious that in the western blots shown basically all Cullins except Cul3 are detected as a mixed population of two species: neddylated and unneddylated. This is as it is to be expected in most cell lines. Any particular reason Cul3 is detected as almost completely in the neddylated form in untreated cells in both THLE2 and KYSE70 (but curiously not HepG2)? Any particular reason for using these types of liver cells in these experiments? What about other types of cancer lines? Some orthogonal methods to fully validate that the inhibitory effect of DI-591 is only on Cul3 while sparing neddylation of the other Cullins would strengthen the claims. The authors are also encouraged to speculate what could give rise to the observed selectivity.

3) The authors provide convincing evidence for the activity of their compounds in cells. However, they do not really show that the observed effect is on-target. To really establish this important validation, ideally the effect of the compounds should be assessed in cells where the endogenous wild-type protein is substituted (knock-in) by a functionally silent but inhibitor-resistant mutant. This may be difficult to design in, given the inhibitor and peptide binding sites overlap, however the binding modes are distinct so it may be possible for example to bump the inhibitor out (thus weakening its binding affinity considerably) by introducing a bulkier hydrophobic residue in the deep hydrophobic hotpot pocket, without affecting much Ubc12 binding. I appreciate this may be a lot of work and outside the scope of the paper, but it could be discussed and considered as future work. While the above would be ideal, a simpler experiment would be to assess inhibitor activity in a target-null cell line i.e. where Dcn1 (and Dcn2 as control) have been knocked out homozygously e.g. using Crispr. The inhibitor effect should be lost in such background, but restored when the target is reintroduced exogenously. These experiments would really provide strong evidence that the chemical probe Cul3-dependent activity is on-target on Dcn1 (and not Dcn2).

4) The Discussion mostly repeats the results. Instead it should provide a broader scholarly view on the implications and impact of the findings to the field, and future directions.

5) Top paragraph of the Discussion (pg. 17, lines 2-6). Suggest the sentence is moderated. In fact the claim that “Very few potent and specific noncovalent small-molecule modulators targeting UPS components have been reported.” is not entirely accurate. Indeed quite a few inhibitors targeting proteins involved in the UPS have been described, for example MDM2 inhibitor Nutlin (Vassilev, L. T. et al. In vivo activation of the p53 pathway by small-molecule antagonists of MDM2. Science 303, 844–848 (2004).), VHL inhibitor VH298 (Frost, J. et al. Potent and selective chemical probe of hypoxic signalling downstream of HIF- α hydroxylation via VHL inhibition. Nat Commun 7, 13312 (2016)), and inhibitors/antagonists of IAP proteins (Fulda, S. & Vucic, D. Targeting IAP proteins for therapeutic intervention in cancer. Nat Rev Drug Discov 11, 109–124 (2012)) amongst others.

These examples should be mentioned and appropriate referenced.

6) Pg. 16, lines 19-23: "Our DCN1 knockdown experiments further established that DCN1 is more critical for neddylation of cullin 3 than for neddylation of other cullins. It is notable that DI-591 causes both depletion of neddylated cullin 3 and accumulation of un-neddyated cullin 3, whereas knockdown of DCN1 primarily increases the levels of un-neddyated cullin 3, which may be attributed to incomplete knockdown of DCN1." As well as sentence on Pg. 18 line 16-18. The effect as evidenced by the western blot is partial, so the authors are encouraged to tone down the claim and add caveats of the siRNA experiment (as highlighted above)

Other points:

- Figure 2b shows competition FP data, presumably using a fluorescent-labelled ligand at fixed concentration and titrating the inhibitor in dose-response curve. This should be clarified in the main text. In the present form, it is unclear which fluorescent labelled ligand is being used. If it is Compound 46 as assumed by this reviewer, then 46 should be introduced earlier in the narrative to avoid confusion.
- Include omit density map $F_o - F_c$ contoured around the modelled ligand in figure 3. This is important to demonstrate the quality of an unbiased density and the fidelity of the model
- Fig. 4e: have the authors monitored Dcn2 in this experiment? Are any Cullins co-IP'd in these experiments? If Cul3 is selective pulled-down compared to other Cullins that may potentially explain the reason for the inhibitor selectivity.
- Fig 4 panel b includes label "BL-DI-781", presumably for the biotinylated compound, which is not introduced elsewhere. Data for the binding of the biotinylated compound in vitro Vs recombinant proteins should be provided.
- Fig 5b: was the MLN4924 treatment for 24 h not toxic to cells? The authors measured ic_{50} of 50-350 nM for this compound (pg. 16, line 10)
- Fig. 5c: statistical significance from p-value analysis should be provided
- define abbreviation for Dmf (pg. 15, line 18)
- Discussion: "excellent aqueous solubility (>20 mM)." Really above 20 mM?
- Supp. Info: missing pg. numbers, so difficult to reference. Chemical structure of compound 47 (last pg. of SI) shown in blue colour. Any reason?

Reviewer #2 (Remarks to the Author):

Overall perspective: The work conducted within this manuscript is both novel and impactful. It addresses a very important research endeavor within the ubiquitin-proteasome field. The extensive medicinal chemistry research effort has afforded a soluble drug-like small-molecule inhibitor wherein the authors provide sufficient evidence to suggest DI-591 disrupts the DCN1-UBC12 protein-protein interaction. DI-591 can be used as a chemical probe to investigate the role of cullin 3 in normal biological processes and in disease progression.

I recommend publication with the following modifications.

1. The authors describe the process by which proteins become neddylated in a brief and

comprehensive manner. However, it might be nice to have a figure that explains this pathway where the authors may use this pictorial representation to allude to why targeting the DCN1-UBC12 interface can lead to the development of a very useful research tool that can help delineate the biological mechanisms underlying the importance of this labelling system.

2. How were K_i 's for all compounds prior to the discovery of DI-591 obtained? The authors allude to FP experiments using 46 in the experimental section but compound 46 was surely only synthesized after the discovery of DI-591. What fluorescently labelled probe was used in the FP experiments prior to the synthesis and use of 46? Was this a fluorescently labelled peptide?

3. How were K_i 's obtained using GraphPad? Equations for non-linear regression data should be provided.

4. No errors for the K_i 's obtained were indicated, these should be explicitly described and/or outlined within the text, especially so since the authors always refer to the relative potencies between the different analogues generated. Otherwise these comparisons are not statistically relevant.

5. The authors indicate that substrates for NEDD8 in Figure 5 only increase by 2-3 kDa on the immunoblots, however the MW of NEDD8 is about 9 kDa. Why does there only appear to be a 2-3 kDa difference in MW, were these protein lysates separated by gradient SDS-PAGE or on a single % resolving gel?

6. Figure 4b – 10 μ M dose for DI-591 looks like 1 over 0 and BL-DI-781 should be labelled 47 for clarity since this code was not mentioned within the text or figure caption

Minor typographical issues

1. Reference numbers come both before and after punctuation at various points throughout the text.

2. Although they use FP, they say these compounds bind DCN1, should state that these potentially displace the fluorescently labelled peptide/probe

3. Page 6 – m-CF3 – change to m-CF₃

4. Kinetic parameters K_{on} and K_{off} should be lower case k_{on} and k_{off}

5. Page 12 – first paragraph line 9 – 47 should be bold

6. Figure 4 caption – two part e's, first should be part d

7. Page 14 Line 18 – add n to _uclear factor erythroid 2-related factor 2

Reviewer #3 (Remarks to the Author):

Review of A Potent, Cell-Permeable Small-Molecule Inhibitor of the DCN1-UBC12 Protein-Protein Interaction that Selectively Blocks Neddylation of Cullin 3

The authors provide a description of a DCN1 inhibitor that was developed based on the N-Terminal domain of UBC12. They show the compound has nanomolar affinity for DCN1 and DCN2 in vitro experiments. However, in cells they claim that only DCN1 is a target for the drug and furthermore inhibiting DCN1 selectively blocks Cul3 neddylation resulting in the stabilization of the Cul3 substrate NRF2.

While the synthesis of the inhibitor and its ability to block the DCN1-UBC12 interactions seems to be thorough and convincing, more work needs to be performed on the use of the compound in cells before I can recommend publication in Nature Communications.

The specific concerns are listed below:

1) The authors state that the compound is specific for DCN1 in cells however the compound binds in vitro to DCN2 with the same K_d as DCN1. Furthermore all of the residues of DCN1 mentioned in the co-crystal structure are conserved in DCN2. The compound should be able to inhibit DCN2 in cells as well. Their data that suggests that DCN2 isn't affected is based on their work with a single

uncharacterized antibody. We have personal experience that the antibody for DCNL2 does not work and the band at 35 kd is not DCNL2. The authors need to verify the specificity of their antibody to support their claims.

2) The concentrations that were used in cells are 1000 times higher than the K_i determined in *in vitro* experiments. While the authors excluded that DCNs 3-5 are affected by the compound, they failed to check for direct inhibition of the E1. Ubc12 binds to the E1 and its N-terminal domain makes contact. It is plausible that a compound designed from the structure of UBC12 would be able to compete off E2 from the E1. Looking at figure 5A, the 30 nM MLN4924 treatment isn't that dissimilar to 10 μ M treatment with D1-591, and if the authors went below 30 nM for MLN4924 it would probably look closer as well. The authors should perform *in vitro* neddylation reactions with D1-591 to show that there is not inhibition of the E1 enzyme. In addition, cell lysates could be prepared using non-reducing conditions to show that Ubc12-NEDD8 thioester is still formed in cells in the presence of 10 μ M D1-591.

3) If the compound is inhibiting DCN1 it should act as a dominant negative that is trapped on a Cullin. If the compound specifically impairs DCN1-Cul3 activity only, then in the pulldowns with biotinylated compound there should also be Cul3 precipitated but not the other Cullins. Keap1 should also be present.

4) Consider performing a rescue experiment for the siRNA DCN1 experiment in figure 5d.

5) What are the expression levels for the other DCNs in this cell line. If you want to say that DCN1 is specific for Cul3-NRF2 activation and can't be compensated for by the other DCNs then you need to show that the other DCNs are present in the cell type used.

6) The effect of the inhibitor on CUL3 neddylation is convincing in the cell lines used. However, they are rather uncommon cells and it is important to add data for more commonly used lines, such as U2OS, HEK293 and HeLas. It would be hugely informative to see whether the effects can be reproduced in these cell lines or whether these are rather cell type specific. Inclusion of a non-transformed primary cell line would also be valuable. If the compound isn't species specific non-transformed primary MEFs may be the easiest to obtain.

Other points :

6) Too much chemistry in the beginning for a general audience.

7) Their reason to target the UPS was for anti-cancer therapy but they found something that increases the amount of NRF2, which won't be useful for cancer treatment, in particular as constitutive NRF2 expression may be oncogenic for some small cell lung cancers. Consider rewriting the intro to include more broad ideas about why the Cullins would want to be targeted and discuss the limitations.

Typos:

On page 3 line 3 consider rephrasing "via tagging the ubiquitin on the proteins" to "by covalent modification of target proteins with ubiquitin".

On page 11 line 4 V164 should be F164

On page 11 line 9 T181 should be Y181

On page 14 line 18 "uclear" should be nuclear

On page 34 line 4 The figure legend states there is a blot for DCNL3 but there isn't one in the figure.

On page 34 panel (b) the concentration of D1-591 at 10 has been split to two lines

On page 34 panel (d) the μM symbol should be moved in line with the numbers.

On page 35 and throughout. The molecular weight markings should be based on a molecule weight ladder and not the expected size of the protein. Moreover NEDD8 is 8 kd so the difference between the modified and unmodified forms should be more than 2 kd.

Point to point response to comments made by reviewers

Response to the comments of Reviewer #1:

This paper reports solid work leading to the discovery of a novel chemical probe targeting the ubiquitin proteasome system. Starting from a crystal structure solved by the Schulman lab in 2014 (Scott, D. C. et al. Cell 157, 1671–1684 (2014)) of the pentameric complex RBX1-UBC12~NEDD8-CUL1CTD-DCN1, which showed that the E2 UBC12 docks its C-terminal tail into a deep groove of the DCN1 protein, the authors embarked in a systematic and well-executed medicinal chemistry campaign combining structure-guided peptidomimetic approaches yielding compound 44 (DI-591) binding DCN1 with high-affinity (K_i of 10 nM based on a competitive FP assay). The biophysical data to support bona-fide reversible interaction of the compound to its target is robust, employing BLI as well as FP, and showing that the inhibitor binds with similar potency to both Dcn1 and (presumably) paralogous protein Dcn2, but not to Dcn3-5. They also synthesized an enantiomer compound and showed that it binds with much weaker affinity (>500 fold less potent), making it an ideal negative control compound for cellular studies. The authors then solved the crystal structure of the compound bound to Dcn1 using co-crystallization. The structure confirmed the binding mode expected based on the rational design, and revealed some important differences relative to the bound peptide that the authors appropriately describe.

The team goes on to use compound 47, a biotinylated analogue of the most potent inhibitor, in target pull-down experiments from cell lysates. Here the first unexpected result is seen, i.e. that the biotinylated compound pulls down in a specific fashion Dcn1, but it does not pull down Dcn2 at all (Fig.

4b). This result is confirmed using CETSA in both temperature dependent (Fig 4c) and dose-dependent manner (Fig. 4d). co-IP experiments confirmed DI-591 blockade of the Dcn1-Ubc12 protein-protein interaction in cells (fig. 4e). Finally, functional characterization of the chemical probe in three different cell lines – THLE2 liver cell line (Fig. 5), liver cancer HepG2 cells (Supp Fig 4) and KYSE70 esophageal cancer cell line (S. Fig 3) – gave a second unexpected result: that is the inhibitor appear to only block neddylation of Cul3 but not any of the other Cullins tested (Cul1, Cul2, Cul4A/B and Cul5) in a dose-dependent manner. This was surprising given Dcn1 had been involved in Cul1 neddylation. Because blockade of Cul3 neddylation is expected to provide a more limited impact in stabilizing proteasomal substrates, compared to eg blockade of neddylation of all Cullins using MLN4924, the authors elegantly assess the impact of their Dcn1 chemical probe on protein levels and activity of Nrf2 – a known Cul3 substrate via the Cul3-Keap1 CRL activity. They show that Nrf2 protein levels go indeed up in concentration dependent manner, but not levels of non-Cul3 substrates e.g. Cyclin E (a Cul1 substrate) and others (Fig. 5a). This leads to upregulation of downstream Nrf2 target genes such as HO-1 and NQO1 at both mRNA level (Fig. 5c) and protein level, but no change in Nrf2 mRNA level, consistent with the activity being due to post-translational stabilization of Nrf2 via blockade of its ubiquitination and degradation. Finally, the authors show that the Cul3-selectivity of their Dcn1 inhibitor is to a degree consistent with siRNA knockdown of Dcn1. This is a somewhat tricky experiment that may not be as conclusive as expected, because it is known that target inhibition can have very different mechanistic consequences from target removal, which in any case is usually incomplete and requires long treatments with siRNA, leading to potential compensatory effects.

Overall, this is a nice article that discloses a novel and potentially useful

chemical probe for studying the function of Dcn1, and in particular, if further validated, the selective blockade of Cul3 neddylation. It is a first in the field, and a worthy advance, and as such I believe that it would eventually merit publication in a journal of the calibre of Nature Communication. However, in my opinion the work suffers from several shortcomings, and that the evidence as presented does not support fully the claims made – as highlighted below – which warrants these issues to be suitably addressed experimentally in a revised version before the paper is ready for publication.

Major points:

Question 1: The first unexpected result concerns the lack of inhibitor binding to cellular Dcn2. This contradicts the biophysical data in vitro using recombinant proteins, and warrants further investigation. **(1-1)** Firstly, the Dcn2 antibody used should be shown to be siRNA or CRISPR proof, to really validate that it is detecting the expected protein. **(1-2)** The authors should also provide more information on (paralogs?) Dcn2-5, e.g. how they compare / differ relative to Dcn1, and reference any relevant literature. What are the sequence identities for example in the domain where the Ubc12 and inhibitors bind? A figure including bars showing the different lengths and domain arrangements of the five proteins would be helpful. **(1-3)** Ultimately, the reader would be left wondering why does the probe not bind to native Dcn2? Can the author provide additional evidence and speculate on possible reasons?

Response 1-1: Considering the Reviewer's suggestion, we employed DCN2 siRNAs to validate DCN2 antibodies. Using a validated DCN2 antibody, we could not detect the expected full-length DCN2 protein in all the cell lines examined. Instead, the detected DCN2 protein in our study has a MW of 21 kDa, which matches the theoretical MW of the shorter DCN2 splicing isoform (sequence: 1-186) (The UniProt, C. UniProt: the universal protein

knowledgebase. *Nucleic Acids Res* **45**, D158-D169 (2017)). We analyzed the Cul1-RBX1-UBC12-NEDD8-DCN1 complex structure (Scott, D.C., *et al.* Structure of a RING E3 Trapped in Action Reveals Ligation Mechanism for the Ubiquitin-like Protein NEDD8. *Cell* **157**, 1671-1684 (2014).) and found that while the detected DCN2 isoform retains the acetylated UBC12 binding site, it lacks the C-terminus domain, which is required for interaction with the cullin protein. We therefore propose that the short splicing isoform of DCN2 detected in all the cell lines in our study is functionally not equivalent to the DCN1 protein and the cellular target for the effective inhibition of neddylation of cullin 3 by DI-591 is DCN1, but not DCN2.

Response 1-2: Per request for providing more information on DCN2-5, the multiple sequence alignment of DCN1-5 was performed and the result was shown in Supplementary Figure S2 with the binding site residues highlighted to show the differences among them. The strong binding affinities of DI-591 to DCN1 and DCN2 and its weak binding affinity to DCN3-5 proteins are consistent with the fact that the DI-591 binding site residues in DCN1 and DCN2 are identical with the exception of the residue 83 (I83 vs V83), and while DCN3-5 proteins have significant differences in their binding site as compared to DCN1/2 (Supplementary Fig. S2).

Response 1-3: Employing a validated DCN2 antibody, we have demonstrate that DI-591 binds to both cellular DCN1 and cellular DCN2 (Figure 4). In co-immunoprecipitation pulldown experiments, **47** efficiently pulls down both cellular DCN1 and DCN2 proteins in a dose-dependent manner in the cell lysate of the KYSE70 esophageal cancer cell line (**Fig. 4b**), which has high DCN1 expression (**SI, Fig. S5a**). Moreover, in a competitive pull-down assay, DI-591 dose-dependently inhibits the binding of **47** with both cellular DCN1 and DCN2 proteins, whereas DI-591DD has no

effect (**Fig. 4b**). Similar pull-down results were also obtained in the KYSE140 esophageal cancer cell line with a high DCN1 expression (**SI, Fig. S5b**). These results indicate that both biotinylated **47** and DI-591 potentially bind to cellular DCN1 and DCN2.

The cellular thermal shift assay shows cellular DCN1 protein was largely degraded at ≥ 50 °C in KYSE70 cells treated with DMSO or DI-591DD (**Fig. 4c**). The thermal stability of DCN1 protein was clearly enhanced by DI-591 at both 50 °C and 55 °C. Furthermore, DI-591 enhances the stability of DCN1 protein in a dose-dependent manner, whereas DI-591DD has no effect (**Fig. 4d**). The thermal stability of DCN2 is also enhanced by DI-591, although DCN2 is more sensitive to heating than DCN1 (**Fig. 4d, e**). These data suggest that DI-591 engages both DCN1 and DCN2 proteins in cells. Consistently, co-immunoprecipitation (Co-IP) experiments showed that the association of cellular DCN1 and UBC12 proteins was effectively reduced by DI-591 but not by DI-591DD (**Fig. 4f**).

Question 2: The conjugation of Nedd8 to Cullins had been shown to require a ubiquitin-like enzyme cascade involving the Nedd8-activating enzyme E1, the Nedd8-conjugating enzyme E2 (Ubc12), the RING protein Rbx1, and the activator E3 Dcn1, resulting in neddylation of Cul1 at lysine 720 (Kurz et al., 2005 Nature 435, 1257–1261; Pan et al., 2004 Oncogene 23, 1985– 1997). (**2-1**) These two articles provide key discoveries related to the Dcn1 dependent neddylation pathway and should be cited. Further, Schulman et al had shown that RBX1 and Dcn1 function synergistically via a “dual E3” mechanism (Scott, D. C. et al. 2010 Mol. Cell 39, 784–796.), mainly on Cul1 as previously shown, but also on other Cullins. It is therefore quite surprising that the Dcn1 inhibitor appears to affect neddylation of only Cul3 but not the other cullins. (**2-2**) Firstly, the authors are encouraged to provide additional

evidence to validate and support this remarkable discovery. **(2-3)** Is the Cul3 antibody used SiRNA or Crispr proof?

Response 2-1: As reviewer suggested, we have carefully studied these two papers (Kurz et al., 2005 Nature 435, 1257–1261; Pan et al., 2004 Oncogene 23, 1985– 1997) and cited them in our manuscript. The introduction was re-organized accordingly.

Response 2-2: We expanded our study in three commonly used cancer cell lines (Supplementary Fig. S7-S9). The selective inhibition of Cul3 neddylation by DI-591 was observed in all the six cell lines we tested.

Response 2-3: Using specific cullin3 siRNA, we confirmed the reliability of cullin3 antibody (2759, CST) we used in our study as shown in the following figure.

Verification of Cullin3 antibody. KYSE70 and HeLa cell lines were transfected with smart-pool siCullin3 (siCul3) from GE Dharmacon or non-target siControl siRNA (siCTL), or DCN2 siRNA (siDCN2, GE Dharmacon) for 48 h. Cells were harvested and the expression of Cullin3 was examined by western blotting analysis with Cullin3 antibody used for western blotting analysis (#2759, CST). GAPDH was used as a loading control.

Question 3: This reviewer found it curious that in the western blots shown basically all Cullins except Cul3 are detected as a mixed population of two

species: neddylated and unneddylated. This is as it is to be expected in most cell lines. **(3-1)** Any particular reason Cul3 is detected as almost completely in the neddylated form in untreated cells in both THLE2 and KYSE70 (but curiously not HepG2)? **(3-2)** Any particular reason for using these types of liver cells in these experiments? **(3-3)** What about other types of cancer lines? Some orthogonal methods to fully validate that the inhibitory effect of DI-591 is only on Cul3 while sparing neddylation of the other Cullins would strengthen the claims. **(3-4)** The authors are also encouraged to speculate what could give rise to the observed selectivity.

Response 3-1: We expanded our study in three commonly used cancer cell lines (Supplementary Fig. S7-S9). Among the six cell lines we tested, the phenomenon of completely neddylated form of Cul3 only occurs in THLE2 and KYSE70 cell lines, but not in other four cell lines. We have no evidence to conclude what causes this discrepancy among different cell lines.

Considering that KYSE70 cell line expresses a very high level of DCN1 (Supplementary Fig. S5a) and also the essential role of DCN1 in cullin3 neddylation, we speculate that a large amount of DCN1 may continuously mediate cullin3 neddylation at high velocity in this cell line. For THLE2 cells, because the function in antioxidation and detoxification of this liver cell line is heavily regulated by NRF2, there is a possibility that a high level of neddylated form (activated form) of cullin3 may play a role in preventing NRF2 pathway from over activation.

Response 3-2: Because liver cells belong to metabolically very active cell types, whose function is heavily regulated by NRF2, we used immortalized normal liver THLE2 cells and liver cancer HepG2 cell lines to investigate the effect of DCN1 inhibitors on NRF2 pathway.

Response 3-3: We have tested DI-591 in three more cell lines and found that DI-591 has the same profound inhibitory effect on neddylation of cullin3 over other cullins in all these cell lines (Supplementary Fig. S7-S9). These

facts suggest that DCN1 plays an essential role in cullin3 neddylation, and may have only a minor role in the neddylation of other cullins.

Response 3-4: Several factors may contribute to the observed highly selective inhibition of CUL3 neddylation by DI-591. We have discussed this in the discussion section as following:

Although DI-591 can effectively interact with both cellular DCN1 and DCN2 proteins (**Fig. 4** and **Supplementary Fig. S5b**), knock-down of DCN1 leads to effective and selective inhibition of neddylation of cullin 3 (**Figure 5d** and **Supplementary Fig. S12**), whereas efficient knock-down of DCN2 has no effect on neddylation of cullin 3 (**Supplementary Fig. S13**). Using a validated DCN2 antibody, we could not detect the expected full-length DCN2 protein in all the cell lines examined. Instead, the detected DCN2 protein in our study has a MW of 21 kDa, which matches the theoretical MW of the shorter DCN2 splicing isoform (sequence: 1-186) (The UniProt, C. UniProt: the universal protein knowledgebase. *Nucleic Acids Res* **45**, D158-D169 (2017)). We analyzed the Cul1-RBX1-UBC12-NEDD8-DCN1 complex structure (Scott, D.C., *et al.* Structure of a RING E3 Trapped in Action Reveals Ligation Mechanism for the Ubiquitin-like Protein NEDD8. *Cell* **157**, 1671-1684 (2014).) and found that while the detected DCN2 isoform retains the acetylated UBC12 binding site, it lacks the C-terminus domain, which is required for interaction with the cullin protein. We therefore proposed that the short splicing isoform of DCN2 detected in all the cell lines in our study is functionally not equivalent to the DCN1 protein and the cellular target for the effective inhibition of neddylation of cullin 3 by DI-591 is DCN1, but not DCN2.

Although cells have three other DCN proteins (DCN3-5), DI-591 fails to bind to DCN3-5 proteins. Our analysis (**Supplementary Fig. S2**) showed that while DCN1 and DCN2 proteins have identical amino acid residues

forming the UBC12 binding site, DCN3-5 proteins have major differences from DCN1 and DCN2 in their amino acid residues in the same region. Although DCN3-5 proteins also bind to UBC12, our data ((**Fig. 5a** and **Supplementary Fig. S6-S11**) suggested that once the interaction of DCN1 with UCB12 is blocked by DI-591, DCN3-5 cannot substitute DCN1 to maintain the neddylation activity for the cullin 3 complex.

As previously shown by Monda et al (Monda, J.K., *et al.* Structural Conservation of Distinctive N-terminal Acetylation-Dependent Interactions across a Family of Mammalian NEDD8 Ligation Enzymes. *Structure* **21**, 42-53 (2013).) in *in vitro* assays, DCN1 promotes UBC12-associated neddylation of CUL1, 2, 3, and 4. However, DCN1 binds to CUL3 more potently than to CUL1, CUL2, CUL4A and CUL4B. This difference may partially account for our observed remarkable selectivity for DI-519 in inhibition of neddylation of cullin 3 over other cullins. The underlying precise mechanisms for the remarkable selectivity of neddylation of cullin 3 over other cullin members clearly deserve further investigation.

Question 4: The authors provide convincing evidence for the activity of their compounds in cells. However, they do not really show that the observed effect is on-target. To really establish this important validation, ideally the effect of the compounds should be assessed in cells where the endogenous wild-type protein is substituted (knock-in) by a functionally silent but inhibitor-resistant mutant. This may be difficult to design in, given the inhibitor and peptide binding sites overlap, however the binding modes are distinct so it may be possible for example to bump the inhibitor out (thus weakening its binding affinity considerably) by introducing a bulkier hydrophobic residue in the deep hydrophobic hotpot pocket, without affecting much Ubc12 binding. I appreciate this may be a lot of work and

outside the scope of the paper, but it could be discussed and considered as future work.

While the above would be ideal, a simpler experiment would be to assess inhibitor activity in a target-null cell line i.e. where Dcn1 (and Dcn2 as control) have been knocked out homozygously e.g. using Crispr. The inhibitor effect should be lost in such background, but restored when the target is reintroduced exogenously. These experiments would really provide strong evidence that the chemical probe Cul3-dependent activity is on-target on Dcn1 (and not Dcn2).

Response 4: This is an excellent recommendation. We believe solving this issue will provide conclusive evidence for the question related to on-target effect of DCN1 inhibitors. However, due to the time constraints, we could not conduct the suggested experiment. We believe that our data have provided convincing support that blocking of the DCN1-UBC12 protein-protein interaction selectively inhibits neddylation of cullin3 over other cullins.

The on-target activity could be evidenced in part by the negative control compound DI-591DD. To evaluate the on-target selectivity of DI-591, we have designed and made a negative control compound DI-591DD, the enantiomer of DI-591. DI-591DD binds to DCN1 with a K_i value of $>6 \mu\text{M}$ and is >500 times less potent than DI-591. DI-591 is able to disrupts the DCN1-UBC12 interaction in cells and convert cellular cullin 3 into exclusively unmodified form without affecting the neddylation status of other cullins, while DI-591DD has little effect. Because DI-591 and DI-591DD are enantiomers and have the same physicochemical properties, these experiments provide evidence that the chemical probe Cul3-dependent activity is on-target on Dcn1.

Question 5: The Discussion mostly repeats the results. Instead it should

provide a broader scholarly view on the implications and impact of the findings to the field, and future directions.

Response 5: We have extensively revised the discussion section based on this nice advice, such as adding the discussion on the possible mechanism of DI-591's selectivity on the cullin 3 neddylation, and the potential implication and impact of the findings in the NRF2 related human diseases.

Question 6: Top paragraph of the Discussion (pg. 17, lines 2-6). Suggest the sentence is moderated. In fact the claim that "Very few potent and specific noncovalent small-molecule modulators targeting UPS components have been reported." is not entirely accurate. Indeed quite a few inhibitors targeting proteins involved in the UPS have been described, for example MDM2 inhibitor Nutlin (Vassilev, L. T. et al. In vivo activation of the p53 pathway by small-molecule antagonists of MDM2. *Science* 303, 844–848 (2004).), VHL inhibitor VH298 (Frost, J. et al. Potent and selective chemical probe of hypoxic signalling downstream of HIF- α hydroxylation via VHL inhibition. *Nat Commun* 7, 13312 (2016)), and inhibitors/antagonists of IAP proteins (Fulda, S. & Vucic, D. Targeting IAP proteins for therapeutic intervention in cancer. *Nat Rev Drug Discov* 11, 109–124 (2012)) amongst others. These examples should be mentioned and appropriate referenced.

Response 6: We have revised this part according to reviewer's kind suggestion.

Question 7: Pg. 16, lines 19-23: "Our DCN1 knockdown experiments further established that DCN1 is more critical for neddylation of cullin 3 than

for neddylation of other cullins. It is notable that DI-591 causes both depletion of neddylated cullin 3 and accumulation of un-neddylated cullin 3, whereas knockdown of DCN1 primarily increases the levels of un-neddylated cullin 3, which may be attributed to incomplete knockdown of DCN1.” As well as sentence on Pg. 18 line 16-18. The effect as evidenced by the western blot is partial, so the authors are encouraged to tone down the claim and add caveats of the siRNA experiment (as highlighted above)

Response 7: We have revised this part according to reviewer’s suggestion.

Other points:

– Figure 2b shows competition FP data, presumably using a fluorescent-labelled ligand at fixed concentration and titrating the inhibitor in dose-response curve. This should be clarified in the main text. In the present form, it is unclear which fluorescent labelled ligand is being used. If it is Compound 46 as assumed by this reviewer, then 46 should be introduced earlier in the narrative to avoid confusion.

Response: All the IC_{50} and the K_i values in this manuscript were obtained by using fluorescent labelled ligand 46. To clarify this, Figure 2 has been rearranged and the main text has been revised accordingly.

– Include omit density map F_o-F_c contoured around the modelled ligand in figure 3. This is important to demonstrate the quality of an unbiased density and the fidelity of the model

Response: The difference electron density map calculated prior to including **DI-591** in the structural refinement is added as Supplementary Figure S4.

– Question about Fig. 4e: have the authors monitored Dcn2 in this

experiment? Are any Cullins co-IP'd in these experiments? If Cul3 is selective pulled-down compared to other Cullins that may potentially explain the reason for the inhibitor selectivity.

Response: We have repeated pulldown experiments with special attention to cullin3 and cullin1. We could not pulldown both cullins with compound **47**.

– Fig 4 panel b includes label “BL-DI-781”, presumably for the biotinylated compound, which is not introduced elsewhere. Data for the binding of the biotinylated compound in vitro Vs recombinant proteins should be provided.

Response: “BL-DI-781” should be **47** and has been corrected. Data for the binding of biotinylated compound **47** in vitro vs recombinant proteins are shown in the Fig. 4 panel a.

– Fig 5b: was the MLN4924 treatment for 24 h not toxic to cells? The authors measured IC50 of 50-350 nM for this compound (pg. 16, line 10)

Response: We have tested MLN4924 for its effect on cell viability with 24 hr treatment time in six cell lines we used in our study and included the new data in Supplementary Fig. S14.

– Fig. 5c: statistical significance from p-value analysis should be provided

Response: p-Value analysis was added.

– define abbreviation for Dmf (pg. 15, line 18)

Response: DMF has been changed to “dimethyl fumarate”.

- Discussion: "excellent aqueous solubility (>20 mM)." Really above 20 mM?

Response: The solubility of the compound was tested as shown in the Supplementary Figure S1.

- Supp. Info: missing pg. numbers, so difficult to reference. Chemical structure of compound 47 (last pg. of SI) shown in blue colour. Any reason?

Response: Page numbers of supplementary information were added. The color of chemical structure of compound 47 was changed to black.

Response to the comments of Reviewer #2:

Reviewer #2 (Remarks to the Author):

Overall perspective: The work conducted within this manuscript is both novel and impactful. It addresses a very important research endeavor within the ubiquitin-proteasome field. The extensive medicinal chemistry research effort has afforded a soluble drug-like small-molecule inhibitor wherein the authors provide sufficient evidence to suggest DI-591 disrupts the DCN1-UBC12 protein-protein interaction. DI-591 can be used as a chemical probe to investigate the role of cullin 3 in normal biological processes and in disease progression.

I recommend publication with the following modifications.

1. The authors describe the process by which proteins become neddylated in a brief and comprehensive manner. However, it might be nice to have a figure that explains this pathway where the authors may use this pictorial representation to allude to why targeting the DCN1-UBC12 interface can lead to the development of a very useful research tool that can help delineate the biological mechanisms underlying the importance of this labelling system.

Response 1: It is a nice suggestion. We have now included a **Figure 6** to illustrate the major findings of our study.

2. How were Ki's for all compounds prior to the discovery of DI-591 obtained? The authors allude to FP experiments using 46 in the experimental section but compound 46 was surely only synthesized after the discovery of DI-591. What fluorescently labelled probe was used in the FP experiments

prior to the synthesis and use of 46? Was this a fluorescently labelled peptide?

Response 2: All the IC_{50} and the K_i values in this manuscript were obtained by using fluorescent labelled ligand 46. Prior to the synthesis of 46, a fluorescently labelled peptide was used as tracer in the FP experiment. After 46 was synthesized, the binding affinity of all the compounds were re-tested to obtain the K_i in the FP assay using **46** as tracer. This was described in the main context.

3. How were K_i 's obtained using GraphPad? Equations for non-linear regression data should be provided.

Response 3: From the competitive FP assays, IC_{50} values were obtained using the dose response inhibition equation (four parameters, variable slopes) included in the Prism/GraphPad software. K_i values were calculated using methods described before (Nikolovska-Coleska Z, Wang R, Fang X, Pan H, Tomita Y, Li P, Roller PP, Krajewski K, Saito NG, Stuckey JA, Wang S. Development and optimization of a binding assay for the XIAP BIR3 domain using fluorescence polarization. *Anal Biochem.* **2004**, 332, 261-73). The citation has been added.

4. No errors for the K_i 's obtained were indicated, these should be explicitly described and/or outlined within the text, especially so since the authors always refer to the relative potencies between the different analogues generated. Otherwise these comparisons are not statistically relevant.

Response 4: All the compounds were tested at least three times and all the K_i values have standard deviation. The standard deviations have been added in Figure 1.

5. The authors indicate that substrates for NEDD8 in Figure 5 only increase by 2-3 kDa on the immunoblots, however the MW of NEDD8 is about 9 kDa. Why does there only appear to be a 2-3 kDa difference in MW, were these protein lysates separated by gradient SDS-PAGE or on a single % resolving gel?

Response 5: The label for the molecule weight marker was corrected.

6. Figure 4b – 10 uM dose for DI-591 looks like 1 over 0 and BL-DI-781 should be labelled 47 for clarity since this code was not mentioned within the text or figure caption

Response 6: We have revised the Figure legend accordingly.

Minor typographical issues

1. Reference numbers come both before and after punctuation at various points throughout the text.

Response 1: Reference numbers have been made consistent to be before the punctuation.

2. Although they use FP, they say these compounds bind DCN1, should state that these potentially displace the fluorescently labelled peptide/probe.

Response 2: Yes, it is a more accurate description. We have also tested our compounds using Bio-Layer Interferometry (BLI) method which show the direct binding of the compound to DCN1. The binding results from these two different assays are consistent; therefore, we describe the compounds bind to DCN1.

3. Page 6 – m-CF3 – change to m-CF₃.

Repose 3: Revised accordingly.

4. Kinetic parameters K_{on} and K_{off} should be lower case k_{on} and k_{off}

Repose 4: Revised accordingly.

5. Page 12 – first paragraph line 9 – 47 should be bold

Repose 5: Revised accordingly.

6. Figure 4 caption – two part e's, first should be part d

Repose 6: Revised accordingly.

7. Page 14 Line 18 – add n to nuclear factor erythroid 2-related factor 2

Repose 7: Revised accordingly.

Response to the comments of Reviewer #3:

Review of A Potent, Cell-Permeable Small-Molecule Inhibitor of the DCN1-UBC12 Protein-Protein Interaction that Selectively Blocks Neddylation of Cullin 3

The authors provide a description of a DCN1 inhibitor that was developed based on the N-Terminal domain of UBC12. They show the compound has nanomolar affinity for DCN1 and DCN2 in vitro experiments. However, in cells they claim that only DCN1 is a target for the drug and furthermore inhibiting DCN1 selectively blocks Cul3 neddylation resulting in the stabilization of the Cul3 substrate NRF2.

While the synthesis of the inhibitor and its ability to block the DCN1-UBC12 interactions seems to be thorough and convincing, more work needs to be performed on the use of the compound in cells before I can recommend publication in Nature Communications.

The specific concerns are listed below:

Question 1: The authors state that the compound is specific for DCN1 in cells however the compound binds in vitro to DCN2 with the same K_d as DCN1. Furthermore all of the residues of DCN1 mentioned in the co-crystal structure are conserved in DCN2. The compound should be able to inhibit DCN2 in cells as well. Their data that suggests that DCN2 isn't affected is based on their work with a single uncharacterized antibody. We have personal experience that the antibody for DCNL2 does not work and the band at 35 kd is not DCNL2. The authors need to verify the specificity of their antibody to support their claims.

Response 1: This is a critical and excellent question. We are very grateful for the valuable knowledge about DCN2 antibody obtained from reviewer's own experience.

Considering the Reviewer's suggestion, we employed siRNAs against DCN2 to validate DCN2 antibodies. Using a validated DCN2 antibody, we could not detect the expected full-length DCN2 protein in all the cell lines examined. Instead, the detected DCN2 protein in our study has a MW of 21 kDa, which matches the theoretical MW of the shorter DCN2 splicing isoform (sequence: 1-186) (The UniProt, C. UniProt: the universal protein knowledgebase. *Nucleic Acids Res* **45**, D158-D169 (2017)). We analyzed the Cul1-RBX1-UBC12-NEDD8-DCN1 complex structure (Scott, D.C., *et al.* Structure of a RING E3 Trapped in Action Reveals Ligation Mechanism for the Ubiquitin-like Protein NEDD8. *Cell* **157**, 1671-1684 (2014).) and found that while the detected DCN2 isoform retains the acetylated UBC12 binding site, it lacks the C-terminus domain, which is required for interaction with the cullin protein. We therefore propose that the short splicing isoform of DCN2 detected in all the cell lines in our study is functionally not equivalent to the DCN1 protein and the cellular target for the effective inhibition of neddylation of cullin 3 by DI-591 is DCN1, but not DCN2.

Question 2: (2-1) The concentrations that were used in cells are 1000 times higher than the K_i determined in in vitro experiments. While the authors excluded that DCNs 3-5 are affected by the compound, they failed to check for direct inhibition of the E1. Ubc12 binds to the E1 and its N-terminal domain makes contact. It is plausible that a compound designed from the structure of UBC12 would be able to compete off E2 from the E1. Looking at figure 5A, the 30 nM MLN4924 treatment isn't that dissimilar to 10 μ M treatment with D1-591, and if the authors went below 30 nM for MLN4924 it would probably look closer as well. **(2-2)** The authors should perform in vitro

neddylation reactions with D1-591 to show that there is not inhibition of the E1 enzyme. In addition, cell lysates could be prepared using non-reducing conditions to show that Ubc12-NEDD8 thioester is still formed in cells in the presence of 10 μ M D1-591.

Response 2-1: Our cell thermal shift assay showed that DI-591 stabilizes DCN1 protein at concentrations as low as 0.3 μ M, which is consistent with its ability to inhibit neddylation of cullin 3 in cells.

Typically for a small-molecule inhibitor, a 10-fold higher concentration is needed to achieve its biological activity in cellular settings, than its biochemical binding affinity to the recombinant protein.

Response 2-2: Following the suggestion of reviewer, an *in vitro* Ubc12~Nedd8 thio ester forming assay was developed. In this assay, our data showed that while MLN4924 effectively inhibits the UBC12 neddylation, DI-591 has no such effect at concentration up to 100 μ M (Supplementary Fig. S3).

Question 3: If the compound is inhibiting DCN1 it should act as a dominant negative that is trapped on a Cullin. If the compound specifically impairs DCN1-Cul3 activity only, then in the pulldowns with biotinylated compound there should also be Cul3 precipitated but not the other Cullins. Keap1 should also be present.

Response 3: We have repeated pulldown experiments with special attention to cullin3 and cullin1. We could not pulldown both cullins with compound **47**.

We propose that that once the interaction between DCN1 and UBC12 proteins is disrupted by DI-591 (or by compound **47**), the interaction between cullin3 (or cullin 1) with DCN1 becomes unstable due to weak

interactions. Therefore, biotinylated compound 47 only pulls down DCN1 but not the DCN1 associated larger complex.

Question 4: Consider performing a rescue experiment for the siRNA DCN1 experiment in figure 5d.

Response 4: This is an excellent recommendation. However, due to the time constraints, we could not conduct the suggested experiment. However, we believe that our data have provided convincing support that blocking of the DCN1-UBC12 protein-protein interaction selectively inhibits neddylation of cullin3 over other cullins.

The on-target activity could be evidenced in part by the negative control compound DI-591DD. To evaluate the on-target selectivity of DI-591, we have designed and made a negative control compound DI-591DD, the enantiomer of DI-591. DI-591DD binds to DCN1 with a K_i value of $>6 \mu\text{M}$ and is >500 times less potent than DI-591. DI-591 is able to disrupt the DCN1-UBC12 interaction in cells and convert cellular cullin 3 into exclusively unmodified form without affecting the neddylation status of other cullins, while DI-591DD has little effect. Because DI-591 and DI-591DD are enantiomers and have the same physicochemical properties, these experiments provide evidence that the chemical probe Cul3-dependent activity is on-target on Dcn1.

Question 5: What are the expression levels for the other DCNs in this cell line. If you want to say that DCN1 is specific for Cul3-NRF2 activation and can't be compensated for by the other DCNs then you need to show that the other DCNs are present in the cell type used.

Response 5: We have run western blotting for the six cell lines we use in the paper and examined the level of all DCN family members (**Supplementary Figure S5a**). Our data show that all DCNs are expressed in all these cell lines, with some variations.

Question 6: The effect of the inhibitor on CUL3 neddylation is convincing in the cell lines used. However, they are rather uncommon cells and it is important to add data for more commonly used lines, such as U2OS, HEK293 and HeLas. It would be hugely informative to see whether the effects can be reproduced in these cell lines or whether these are rather cell type specific. Inclusion of a non-transformed primary cell line would also be valuable. If the compound isn't species specific non-transformed primary MEFs may be the easiest to obtain.

Response 6: As suggested by reviewer, we have extended western blotting analysis in MDA-MB-231, U2OS and HeLa, three very commonly used cancer cell lines in the field of cancer research. Consistent with our previous findings in the KYSE70, THLE2 and HepG2 cell lines, we found that DI-591 had much more profound inhibitory effect on the neddylation of cullin3 over other cullins in these three cell lines. We have included these findings in our manuscript (Supplementary Fig. S7-S9).

Other points :

6) Too much chemistry in the beginning for a general audience.

Response 6: For this paper, we have two major components: (i) structure-based discovery of DI-591 as a potent, druglike and cell-permeable small-molecule inhibitor and (ii) extensive evaluation of DI-591 in biochemical assays and in cells. We believe that the first part is essential and will be appreciated by the large scientific community, particularly in the area of chemical biology and drug discovery research. Accordingly, we have kept the first part largely intact in the revised manuscript.

7) Their reason to target the UPS was for anti-cancer therapy but they found something that increases the amount of NRF2, which won't be useful for cancer treatment, in particular as constitutive NRF2 expression may be oncogenic for some small cell lung cancers. Consider rewriting the intro to include more broad ideas about why the Cullins would want to be targeted and discuss the limitations.

Response 7: It is very nice suggestion and we have rewritten the introduction.

Typos:

On page 3 line 3 consider rephrasing "via tagging the ubiquitin on the proteins" to "by covalent modification of target proteins with ubiquitin".

Response: Revised accordingly.

On page 11 line 4 V164 should be F164

Response: Revised accordingly.

On page 11 line 9 T181 should be Y181

Response: Revised accordingly.

On page 14 line 18 "uclear" should be nuclear

Response: Revised accordingly.

On page 34 line 4 The figure legend states there is a blot for DCNL3 but

there isn't one in the figure.

Response: Revised accordingly.

On page 34 panel (b) the concentration of D1-591 at 10 has been split to two lines

Response: Revised accordingly.

On page 34 panel (d) the μM symbol should be moved in line with the numbers.

Response: Revised accordingly.

On page 35 and throughout. The molecular weight markings should be based on a molecule weight ladder and not the expected size of the protein. Moreover NEDD8 is 8 kd so the difference between the modified and unmodified forms should be more than 2 kd.

Response: The labels for the molecule weight markers were corrected.

REVIEWERS' COMMENTS:

Reviewer #1 (Remarks to the Author):

The authors have done a very good, thorough job at revising their manuscript based on the reviewers' comments. Specifically, I am happy with the changes implemented to address the points raised by this reviewer. As a result, the article is much improved, and the claims have been adequately revised based on the supporting data provided. I am therefore recommending publication in this current form. There are still a couple minor issues that need fixing before the article is ready:

1) my point about statistical significant for Fig. 5c has not really been addressed. The authors have responded: "p-Value analysis was added" (pg. 13 in the rebuttal letter); however I cannot see any stars in the graphs nor p-value information in the legend.

2) the authors now nicely illustrate how only one residue is different in the binding site between Dcn1 and Dcn2 (specified as "residue 83, I83 v V83" in the rebuttal, and labeled as I83 in Fig. 1a-b). However the numbering is 88 in the main text (pg. 10, line 19). This should be checked carefully and fixed.

3) Figure 4 legend, pg. 35 line 5: "Cell lysates of KYSE140 cell line were treated...": presumably this should be KYSE70 here? The data from the KYSE140 cell line is stated as shown in Supp. Fig. S5, panel b!

Reviewer #2 (Remarks to the Author):

The authors have presented a revised manuscript that addresses all the issues raised by myself and the other reviewers. I believe the manuscript is publishable as it stands in Nature Communications.

Reviewer #3 (Remarks to the Author):

I am happy with the revisions that were performed by the authors and can now recommend publications.

I would, however, like to suggest that the authors add a short sentence in the discussion acknowledging that in a setting where full length DCN2 is expressed, the function of this isoform will likely be equally inhibited than that of DCN1 (this is in reference to my previous suggestions 1).

Also, while it would have been very nice to see a rescue for the DCN1 siRNA experiment (suggestion 4), I no longer deem it absolutely necessary for publication, given the very convincing evidence presented in the remainder of the paper.

I think this is an excellent piece of work and congratulate the authors on their achievement.

Point-by-point response to comments from reviewers

Response to Reviewer #1 comments :

The authors have done a very good, thorough job at revising their manuscript based on the reviewers' comments. Specifically, I am happy with the changes implemented to address the points raised by this reviewer. As a result, the article is much improved, and the claims have been adequately revised based on the supporting data provided. I am therefore recommending publication in this current form. There are still a couple minor issues that need fixing before the article is ready:

1) my point about statistical significant for Fig. 5c has not really been addressed. The authors have responded: "p-Value analysis was added" (pg. 13 in the rebuttal letter); however I cannot see any stars in the graphs nor p-value information in the legend.

Response: The stars and p-values for Fig. 5c have been included.

2) the authors now nicely illustrate how only one residue is different in the binding site between Dcn1 and Dcn2 (specified as "residue 83, I83 v V83" in the rebuttal, and labeled as I83 in Fig. 1a-b). However the numbering is 88 in the main text (pg. 10, line 19). This should be checked carefully and fixed.

Response: Thank you for pointing out this error. I83 is the correct number for the residue. This has been corrected in the main text.

3) Figure 4 legend, pg. 35 line 5: "Cell lysates of KYSE140 cell line were treated...": presumably this should be KYSE70 here? The data from the KYSE140 cell line is stated as shown in Supp. Fig. S5, panel b!

Response: Thank you for pointing out this error. We corrected it.

Reviewer #2 (Remarks to the Author):

The authors have presented a revised manuscript that addresses all the issues raised by myself and the other reviewers. I believe the manuscript is publishable as it stands in Nature Communications.

Response to Reviewer #3 comments:

I am happy with the revisions that were performed by the authors and can now recommend publications.

I would, however, like to suggest that the authors add a short sentence in the discussion acknowledging that in a setting where full length DCN2 is expressed, the function of this isoform will likely be equally inhibited than that of DCN1 (this is in reference to my previous suggestions 1).

Also, while it would have been very nice to see a rescue for the DCN1 siRNA experiment (suggestion 4), I no longer deem it absolutely necessary for publication, given the very convincing evidence presented in the remainder of the paper.

I think this is an excellent piece of work and congratulate the authors on their achievement.

Response: We have added this sentence to the discussion: “However, DI-591 should inhibit the function of full length DCN2 protein in cells where this isoform is expressed.” .